# Improved Regret Bounds for Linear Adversarial MDPs via Linear Optimization

**Fang Kong**[*]                                                                                    *fangkong@sjtu.edu.cn*
*Shanghai Jiao Tong University*

**Xiangcheng Zhang**[*]                                                              *xc-zhang21@mails.tsinghua.edu.cn*
*Tsinghua University*

**Baoxiang Wang**                                                                      *bxiangwang@cuhk.edu.cn*
*The Chinese University of Hong Kong, Shenzhen*

**Shuai Li**[†]                                                                                     *shuaili8@sjtu.edu.cn*
*Shanghai Jiao Tong University*

Reviewed on OpenReview: <https://openreview.net/forum?id=KcmWZSk53y>

## Abstract

Learning Markov decision processes (MDP) in an adversarial environment has been a challenging problem. The problem becomes even more challenging with function approximation since the underlying structure of the loss function and transition kernel are especially hard to estimate in a varying environment. In fact, the state-of-the-art results for linear adversarial MDP achieve a regret of $\tilde{\mathcal{O}}(K^{6/7})$ ($K$ denotes the number of episodes), which admits a large room for improvement. In this paper, we propose a novel explore-exploit algorithm framework and investigate the problem with a new view, which reduces linear MDP into linear optimization by subtly setting the feature maps of the bandit arms of linear optimization. This new technique, under an exploratory assumption, yields an improved bound of $\tilde{\mathcal{O}}(K^{4/5})$ for linear adversarial MDP without access to a transition simulator. The new view could be of independent interest for solving other MDP problems that possess a linear structure.

## 1 Introduction

Reinforcement learning (RL) describes the interaction between a learning agent and an unknown environment, where the agent aims to maximize the cumulative reward through trial and error (Sutton and Barto, 2018). It has achieved great success in many real applications, such as games (Mnih et al., 2013; Silver et al., 2016), robotics (Kober et al., 2013; Lillicrap et al., 2015), autonomous driving (Kiran et al., 2021) and recommendation systems (Afsar et al., 2022; Lin et al., 2021). The interaction in RL is commonly portrayed by Markov decision processes (MDP). Most works study the stochastic setting, where the reward is sampled from a fixed distribution (Azar et al., 2017; Jin et al., 2018; Simchowitz and Jamieson, 2019; Yang et al., 2021). RL in real applications is in general more challenging than the stochastic setting, as the environment could be non-stationary and the reward function could be adaptive towards the agent's policy. For example, a scheduling algorithm will be deployed to self-interested parties, and recommendation algorithms will face strategic users.

To design robust algorithms that work under non-stationary environments, a line of works focuses on the adversarial setting, where the reward function could be arbitrarily chosen by an adversary (Yu et al., 2009;

---

[*]Equal contribution.
[†]Corresponding author.

Rosenberg and Mansour, 2019; Jin et al., 2020a; Chen et al., 2021; Luo et al., 2021a). Many works in adversarial MDPs optimize the policy by learning the value function with a tabular representation. In this case, both their computation complexity and their regret bounds depend on the state space and action space sizes. In real applications however, the state and action spaces could be exponentially large or even infinite, such as in the game of Go and in robotics. Such cost of computation and the performance are then inadequate.

To cope with the curse of dimensionality, function approximation methods are widely deployed to approximate the value functions with learnable structures. Great empirical success has proved its efficacy in a wide range of areas. Despite this, theoretical understandings of MDP with general function approximation are yet to be available. As an essential step towards understanding function approximation, linear MDP has been an important setting and has received significant attention from the community. It presumes that the transition and reward functions in MDP follow a linear structure with respect to a known feature (Jin et al., 2020b; He et al., 2021; Hu et al., 2022). The stochastic setting in linear MDP has been well studied and near-optimal results are available (Jin et al., 2020b; Hu et al., 2022). The adversarial setting in linear MDP is much more challenging since the underlying linear parameters of the loss function and transition kernel are especially hard to estimate while having to adapt to a varying environment.

The research on linear adversarial MDPs remains open. Early works proposed algorithms when the transition function is known (Neu and Olkhovskaya, 2021). Several recent works explore the problem without a known transition function and derive policy optimization algorithms with the state-of-the-art regret of $\tilde{\mathcal{O}}\left(K^{6/7}\right)$ (Luo et al., 2021a; Dai et al., 2023; Sherman et al., 2023; Lancewicki et al., 2023). While the optimal regret in tabular MDPs is of order $\tilde{\mathcal{O}}\left(K^{1/2}\right)$ (Jin et al., 2020a), the regret upper bounds available for linear adversarial MDPs seem to admit a large room for improvement.

In this paper, we investigate linear adversarial MDPs with unknown transitions. We propose a new view that transforms linear MDP to a linear optimization problem by regarding the expected state-action visitation features in MDP as the feature of arms in linear optimization problem. Our algorithm constitutes an exploration and exploitation phase. In the exploration phase, we estimate the feature visitations of the policies in the constructed policy set using reward-free estimation techniques. Then, in the exploitation phase, we operate on a set of policies and optimize the probability distribution of which policy to execute. By carefully balancing the suboptimality in policy construction, the suboptimality in feature visitation estimation, and the suboptimality in policy execution, we deduce new analyses of the problem which achieves the first $\tilde{\mathcal{O}}\left(K^{4/5}\right)$ regret bound for linear adversarial MDPs without a simulator. Let $d$ be the feature dimension and $H$ be the length of each episode. Details of our contributions are as follows.

- With an exploratory assumption (Assumption 1), we obtain an $\tilde{\mathcal{O}}(d^{7/5}H^{12/5}K^{4/5})$ regret upper bound for linear adversarial MDP. As compared in Table 1, this is the first regret bound that achieves $\tilde{\mathcal{O}}(K^{4/5})$ order when a simulator of the transition is not provided. We also want to note that our exploratory assumption that only ensures the MDP is learnable is much weaker than previous works (Neu and Olkhovskaya, 2021; Luo et al., 2021a). Under this weaker exploratory assumption, our result achieves a significant improvement over the $\tilde{\mathcal{O}}(K^{6/7})$ regret in Luo et al. (2021a) and also removes the dependence on $\lambda$ in the main order term which is the minimum eigenvalue of the exploratory policy's covariance that can be small.

- In a simpler setting where the agent has access to a simulator, our regret can be further improved to $\tilde{\mathcal{O}}(\sqrt{d^2H^5K})$. This result also removes the dependence on $\lambda$ in the main order of previous works (Neu and Olkhovskaya, 2021; Luo et al., 2021a). Compared with Luo et al. (2021a), our required simulator is also weaker: we only need to have access to the trajectory when given any policy $\pi$, while Luo et al. (2021a) requires the next state $s'$ given any state-action pair $(s, a)$.

- Our results are the first that achieves $\tilde{\mathcal{O}}(K^{1/2})$ high probability-type regret bound for linear adversarial MDP with a simulator and $\tilde{\mathcal{O}}(K^{4/5})$ regret without a simulator when the exploratory assumption holds. Previous works for high probability-type regret only achieve $\tilde{\mathcal{O}}(K^{2/3})$ bound with access to a simulator for the linear-Q setting (Lancewicki et al., 2023). And Dai et al. (2023); Luo et al. (2021a); Sherman et al. (2023) which are based on the policy optimization approach are only able to produce the upper bound for the expected regret.

- Technically, we provide a new tool and simpler analysis for linear MDP problems by exploiting the linear features of the MDP and transforming it into a linear optimization problem by setting the expected feature mappings of different policies as arms. We also provide a refined analysis of the Hedge algorithm where the features are approximated instead of being precisely given. This approach differs from previous policy optimization-based algorithms and could be of independent interest in other problems that possess a linear structure.

Table 1: Comparisons of our results with most related works for linear adversarial MDPs.

| | Simulator[1] | Exploratory Assumption[2] | Regret[3] | Computation complexity |
|---|---|---|---|---|
| Neu and Olkhovskaya (2021) [4] | yes | yes | $\tilde{\mathcal{O}}(\sqrt{K/\lambda})$ | $\mathrm{poly}(|\mathcal{S}|, |\mathcal{A}|, d, H, K)$ |
| Luo et al. (2021a) | yes | yes | $\tilde{\mathcal{O}}(\sqrt{K/\lambda})$ | $(KAH)^{\mathcal{O}(H)}$ [5] |
| | | no | $\tilde{\mathcal{O}}(K^{2/3})$ | |
| | no | yes | $\tilde{\mathcal{O}}\left(K/(\lambda)^{2/3}\right)^{6/7}$ | $\mathrm{poly}(|\mathcal{A}|, d, H, K)$ |
| | | no | $\tilde{\mathcal{O}}(K^{14/15})$ | |
| Dai et al. (2023) | yes | no | $\tilde{\mathcal{O}}(\sqrt{K})$ | $(KAH)^{\mathcal{O}(H)}$ [5] |
| | no | no | $\tilde{\mathcal{O}}(K^{8/9})$ | $\mathrm{poly}(|\mathcal{A}|, d, H, K)$ |
| Sherman et al. (2023) | yes | no | $\tilde{\mathcal{O}}(K^{2/3})$ | $\mathrm{poly}(|\mathcal{A}|, d, H, K)$ |
| | no | no | $\tilde{\mathcal{O}}(K^{6/7})$ | |
| Ours | yes | yes | $\tilde{\mathcal{O}}(\sqrt{K})$ | $K^{\mathcal{O}(dH^2)}$ |
| | no | yes | $\tilde{\mathcal{O}}(K^{4/5}+\mathrm{poly}(1/\lambda))$ | |

[1] The simulator required in this paper is defined in Assumption 2. Note that the simulators adopted in Dai et al. (2023) and Luo et al. (2021a) are stronger that returns the next state $s'$ when given any state action pair $(s, a)$, while Sherman et al. (2023); Neu and Olkhovskaya (2021) and this paper only need the simulator to return a trajectory given a policy. Our simulator can be implied by theirs.

[2] Our exploratory assumption is introduced in Assumption 1. It is worth noting that our assumption on exploration is also much weaker than Neu and Olkhovskaya (2021); Luo et al. (2021a;b). Specifically, our exploratory assumption only ensures the learnability of the MDP while the other works require a policy that can explore the full linear space in all steps. Our assumption can be implied by theirs.

[3] $\lambda$ in the regret represents the minimum eigenvalue induced by a "good" exploratory policy $\pi_0$, which satisfies $\lambda_{\min}(\mathbf{\Lambda}_{\pi_0,h}) \geq \lambda$ for all $h \in [H]$, where $\mathbf{\Lambda}_{\pi_0,h}$ is the covariance of $\pi_0$ at step $h$ (see Assumption 1).

[4] The algorithm of Neu et al. (2010) requires full knowledge of the transition kernel.

[5] The exponential complexity is due to the exponential number of calls to the simulator when constructing the dilated bonus in Luo et al. (2021a) and Dai et al. (2023).

## 2 Related Work

**Linear Stochastic MDPs.** The linear function approximation problem has been studied for a long history (Bradtke and Barto, 1996; Melo and Ribeiro, 2007; Sutton and Barto, 2018; Yang and Wang, 2019). Until recently, Yang and Wang (2020) propose theoretical guarantees for the sample efficiency in the linear MDP setting. However, it assumes that the transition function can be parameterized by a small matrix. In general cases, Jin et al. (2020b) develop the first efficient algorithm LSVI-UCB both in sample and computation complexity. They show that the algorithm achieves $\tilde{\mathcal{O}}(\sqrt{d^3H^3K})$ regret where $d$ is the feature dimension and $H$ is the length of each episode. This result is improved to the optimal order $\tilde{\mathcal{O}}(dH\sqrt{K})$ by Hu et al. (2022) with a tighter concentration analysis. A very recent work (He et al., 2022a) points out a technical error in Hu et al. (2022) and show a nearly minimax result that matches the lower bound $\tilde{\mathcal{O}}(d\sqrt{H^3K})$ in Zhou et al. (2021). All these works are based on UCB-type algorithms. Apart from UCB, the TS-type algorithm has also been proposed for this setting (Zanette et al., 2020). The above results mainly focus on minimax optimality, and it is also attractive to derive an instance-dependent regret bound as it changes in MDPs

with different hardness. He et al. (2021) is the first to provide this type of regret in linear MDP. Using a different proof framework, they show that the LSVI-UCB algorithm can achieve $\tilde{\mathcal{O}}(d^3 H^5 \log K / \Delta)$ where $\Delta$ is the minimum value gap in the episodic MDP.

**Adversarial losses in MDPs**  When the losses at state-action pairs do not follow a fixed distribution, the problem becomes the adversarial MDP. This problem was first studied in the tabular MDP setting. The occupancy measure-based method is one of the most popular approaches to dealing with a potential adversary. For this type of approach, Zimin and Neu (2013) first study the known transition setting and derive regret guarantees $\tilde{\mathcal{O}}(H\sqrt{K})$ and $\tilde{\mathcal{O}}(\sqrt{HSAK})$ for full-information and bandit feedback, respectively. For the more challenging unknown transition setting, Rosenberg and Mansour (2019) also start from the full-information feedback and derive an $\tilde{\mathcal{O}}(HS\sqrt{AK})$ regret. The bandit feedback is recently studied by Jin et al. (2020a), where the regret bound is $\tilde{\mathcal{O}}(HS\sqrt{AK})$. The other line of works (Neu et al., 2010; Shani et al., 2020; Chen et al., 2022; Luo et al., 2021a) is based on policy optimization methods. In the unknown transition and bandit feedback setting, the state-of-the-art result in this line is also an $\tilde{\mathcal{O}}(\sqrt{K})$ order achieved by Luo et al. (2021a;b).

Specifically, a few works focus on the linear adversarial MDP problem. Neu and Olkhovskaya (2021) first study the known transition setting and provide an $\tilde{\mathcal{O}}(\sqrt{K})$ regret with the assumption that an exploratory policy can explore the full linear space. For the general unknown transition case, Luo et al. (2021a;b) discuss four cases on whether a simulator is available and whether the exploratory assumption is satisfied. With the same exploratory assumption as Neu and Olkhovskaya (2021), they show a regret bound $\tilde{\mathcal{O}}(\sqrt{K})$ with a simulator and $\tilde{\mathcal{O}}(K^{6/7})$ otherwise. Very recent two works (Dai et al., 2023; Sherman et al., 2023) further generalize the setting by removing the exploratory assumption. These two works independently provide $\tilde{\mathcal{O}}(K^{8/9})$ and $\tilde{\mathcal{O}}(K^{6/7})$ regret for this setting when no simulator is available. It is worth noting that the above works provide guarantees for the expected regret. For the stronger high-probability regret objective, Lancewicki et al. (2023) provide an $\tilde{\mathcal{O}}(K^{2/3})$ upper bound when a simulator is available.

Linear mixture MDP is another popular linear function approximation model, where the transition is a mixture of linear functions. When considering the adversarial losses, Cai et al. (2020) and He et al. (2022b) study unknown transition but full-information feedback type, in which case the learning agent can observe the loss of all actions in each state. Zhao et al. (2023a) consider the general bandit feedback in this setting and show the regret in this harder environment is also $\tilde{\mathcal{O}}(\sqrt{K})$. Their modeling does not assume the structure of the loss function which introduces the dependence on $S, A$ in the regret. Apart from linear function approximation, Zhao et al. (2023b) also consider the adversarial losses in low-rank MDPs under full-information feedback.

## 3    Preliminaries

In this work, we study the episodic adversarial Markov decision processes (MDP) denoted by $\mathcal{M}(\mathcal{S}, \mathcal{A}, H, \{P_h\}_{h=1}^H, \{\ell_k\}_{k=1}^K)$ where $\mathcal{S}$ is the state space, $\mathcal{A}$ is the action space, $H$ is the horizon of each episode, $P_h : \mathcal{S} \times \mathcal{A} \times \mathcal{S} \to [0, 1]$ is the transition kernel of step $h$ with $P_h(s' \mid s, a)$ representing the transition probability from $s$ to $s'$ by taking action $a$ at step $h$, and $\ell_k$ is the loss function at episode $k$. We denote $\pi_k = \{\pi_{k,h}\}_{h=1}^H$ as the learner's policy at each episode $k$, where $\pi_{k,h}$ is a mapping from each state to a distribution over the action space of step $h$, and let $\pi_{k,h}(a \mid s)$ represent the selecting probability of action $a$ at state $s$ by following policy $\pi_k$ at step $h$.

The learner interacts with the MDP $\mathcal{M}$ for $K$ episodes. At each episode $k = 1, 2, \ldots, K$, the environment (adversary) first chooses the loss function $\ell_k := \{\ell_{k,h}\}_{h=1}^H$ which may be probably based on the history information before episode $k$. The learner simultaneously decides its policy $\pi_k$. For each step $h = 1, 2, \ldots, H$, the learner observes the current state $s_{k,h}$, takes action $a_{k,h}$ based on $\pi_{k,h}$, and observes the loss $\ell_{k,h}(s_{k,h}, a_{k,h})$. The environment will transit to the next state $s_{k,h+1}$ at the end of the step based on the transition kernel $P_h(\cdot \mid s_{k,h}, a_{k,h})$.

The performance of a policy $\pi$ over episode $k$ can be evaluated by its expected cumulative loss, also known as the value function defined as below:

$$V_k^\pi = \mathbb{E}\left[\sum_{h=1}^H \ell_{k,h}(s_{k,h}, a_{k,h})\right],$$

where the expectation is taken from the randomness in the transition and the policy $\pi$. Denote $\pi^* \in \operatorname{argmin}_\pi \sum_{k=1}^K V_k^\pi$ as the global optimal policy that suffers the least expected loss over $K$ episodes. The objective of the learner is to minimize the cumulative regret which is defined as the cumulative difference between the value of the taken policies and that of the global optimal policy $\pi^*$:

$$\operatorname{Reg}(K) = \sum_{k=1}^K \left(V_k^{\pi_k} - V_k^{\pi^*}\right). \tag{1}$$

Linear adversarial MDP denotes an MDP where both the transition kernel and the loss function are linearly depending on a feature mapping. We give a formal definition as follows.

**Definition 1** (Linear MDP with adversarial losses). *The MDP $\mathcal{M}(\mathcal{S}, \mathcal{A}, H, \{P_h\}_{h=1}^H, \{\ell_k\}_{k=1}^K)$ is a linear MDP if there is a known feature mapping $\phi : \mathcal{S} \times \mathcal{A} \to \mathbb{R}^d$ and $d$ unknown signed measures $\mu_{h,1}, \ldots, \mu_{h,d} : \mathcal{S} \to \mathbb{R}$ forming $\mu_h := (\mu_{h,1}, \ldots, \mu_{h,d}) : \mathcal{S} \to \mathbb{R}^d$ such that the transition probability for any $s, a, h$ satisfies*

$$P_h\left(\cdot|s,a\right) = \langle \phi(s,a), \mu_h \rangle.$$

*Further, for any episode $k$ and step $h$, there exists an unknown loss vector $\theta_{k,h} \in \mathbb{R}^d$ such that*

$$\ell_{k,h}(s,a) = \langle \phi(s,a), \theta_{k,h} \rangle.$$

*for all state-action pair $(s,a)$. Without loss of generality, we assume $\|\phi(s,a)\|_2 \leq 1$ for all $s, a$, $\||\mu_h|(\mathcal{S})\|_2 = \left\|\int_{s \in \mathcal{S}} |d\mu_h(s)|\right\|_2 \leq \sqrt{d}$, $\|\theta_{k,h}\|_2 \leq \sqrt{d}$ for any $k, h$, and $\ell_{k,h}(s,a) \leq 1$ for all $k, h, s, a$.*

Given a policy $\pi$, its *feature visitation* at step $h$ is defined as the expected feature mapping this policy encounters at step $h$ following policy $\pi$: $\phi_{\pi,h} = \mathbb{E}_\pi [\phi(s_h, a_h)]$, where $\mathbb{E}_\pi[\cdot]$ denotes taking expectation w.r.t the randomness of state transitions and policy $\pi$. Then the expected loss that the policy $\pi$ receives at step $h$ of episode $k$ can be written as

$$\ell_{k,h}^\pi := \mathbb{E}_\pi\left[\ell_{k,h}\left(s_{k,h}, a_{k,h}\right)\right] = \langle \phi_{\pi,h}, \theta_{k,h} \rangle, \tag{2}$$

and the value of policy $\pi$ can be expressed as

$$V_k^\pi = \sum_{h=1}^H \ell_{k,h} = \sum_{h=1}^H \langle \phi_{\pi,h}, \theta_{k,h} \rangle. \tag{3}$$

For simplicity, we also define $\phi_{\pi,h}(s) = \mathbb{E}_{a \sim \pi_h(\cdot|s)}[\phi(s,a)]$ to represent the expected feature visitation of state $s$ at step $h$ by following $\pi$.

For any policy $\pi$, define $\mathbf{\Lambda}_{\pi,h} := \mathbb{E}_{(s_h,a_h) \sim \pi}\left[\phi(s_h, a_h)\phi(s_h, a_h)^\top\right]$ as the expected covariance of $\pi$ at step $h$. Let $\lambda_{\min,h}^* = \sup_\pi \lambda_{\min}(\mathbf{\Lambda}_{\pi,\mathbf{h}})$, where $\lambda_{\min}(\cdot)$ denotes the smallest eigenvalue of a matrix, and $\lambda_{\min}^* = \min_h \lambda_{\min,h}^*$. Analog to previous works that study the function approximation setting (Neu and Olkhovskaya, 2021; Luo et al., 2021a; Hao et al., 2021; Agarwal et al., 2021), we make the following exploratory assumption to ensure efficiency in exploring the transition dynamics:

**Assumption 1** (Exploratory assumption). $\lambda_{\min}^* > 0$.

When the assumption is reduced to the tabular setting, where $\phi(s,a)$ is a basis vector in $\mathbb{R}^{\mathcal{S} \times \mathcal{A}}$, this assumption becomes $\mu_{\min} := \min_h \max_\pi \min_{s,a} \mu^\pi(s,a) > 0$, where $\mu^\pi(s,a)$ is the probability of visiting the state-action pair $(s,a)$ under the trajectory induced by $\pi$. It simply means that there exists a policy with

positive visitation probability for all state-action pairs, which is standard (Li et al., 2020). In the linear setting, it guarantees that all the directions in $\mathbb{R}^d$ are able to be visited by some policy.

We point out that this assumption is weaker than the exploratory assumptions used in previous works (Neu and Olkhovskaya, 2021; Luo et al., 2021a), in which they assume such exploratory policy $\pi_0$, satisfying $\lambda_{\min}\left(\mathbf{\Lambda}_{\pi_0,h}\right) \geq \lambda_0 > 0$ for all $h \in [H]$, is given as input to the algorithm. We only assume such policies to exist, which only requires the transition kernel of the MDP to satisfy certain constraints, while finding an exploratory policy is an extremely difficult task.

## 4 Algorithm

In this section, we introduce the proposed algorithm 1. At a high level, the algorithm is based on our proposed new view that transforms the linear MDP to a linear optimization problem by regarding the state-action visitation feature as arms. It takes a finite policy class $\Pi$ and the feature visitation estimators $\{\hat{\phi}_{\pi,h} : \pi \in \Pi, h \in [H]\}$ as input and optimizes the probability of selecting each policy in $\Pi$. The acquisition of $\Pi$ and $\{\hat{\phi}_{\pi,h} : \pi \in \Pi, h \in [H]\}$ will be introduced in Section 4.1 and Section 4.2, respectively.

Recall that the loss value $\ell_{k,h}(s,a)$ is an inner product between the feature $\phi(s,a)$ and the loss vector $\theta_{k,h}$. According to this structure, we investigate ridge linear regression to estimate the unknown loss vector. To be specific, in each episode after executing policy $\pi_k$, the observed loss value can be used to estimate the loss vector $\hat{\theta}_{k,h}$ and the value function $\hat{V}_k^\pi$ can be estimated for each policy $\pi$ (line 9). We then adopt an optimistic strategy toward the values of policies and an optimistic estimation of any policy $\pi$'s value (line 10). Based on the optimistic value, the exploitation probability $w(\pi)$ of a policy $\pi$ follows an EXP3-type update rule (line 11). To better explore each dimension of the linear space, the final selecting probability is defined as the weighted combination of the exploitation probability and an exploration probability $g(\pi)$, where the weight $\gamma$ is an input parameter (line 7). Here the exploration probability $g_h = \{g_h(\pi)\}_{\pi \in \Pi}$ is derived by computing the G-optimal design problem to minimize the uncertainty of all policies, i.e.,

$$g_h \in \operatorname*{argmin}_{p \in \Delta(\Pi)} \max_{\pi \in \Pi} \|\phi_{\pi,h}\|^2_{\mathbf{V}_h(p)^{-1}} \ ,$$

where $\mathbf{V}_h(p) = \sum_{\pi \in \Pi} p(\pi)\phi_{\pi,h}\phi_{\pi,h}^\top$. This exploratory probability can ensure the magnitude of $\hat{V}_{k,\pi}$ being bounded. As for the computation complexity, as outlined Lattimore and Szepesvári (2020, Note 3 of Chapter 21), the optimal design can be approximated efficiently within $O(d \log \log d)$ iterations, yielding $g(\pi)$ that satisfies the following inequality:

$$\max_{\pi \in \Pi} \left\|\hat{\phi}_{\pi,h}\right\|^2_{\mathbf{V}_h(g_h)^{-1}} \leq 2 \min_{p \in \Delta(\Pi)} \max_{\pi \in \Pi} \left\|\hat{\phi}_{\pi,h}\right\|^2_{\mathbf{V}_h(p)^{-1}} \leq 2d \,.$$

And the constant approximation factor will not change the order of the regret.

If the input $\hat{\phi}_{\pi,h}$ is the true feature visitation $\phi_{\pi,h}$, we can ensure that the regret of the algorithm compared with the optimal policy in $\Pi$ can be upper bounded. Now it suffices to bound the additional regret caused by the sub-optimality between the optimal policy in $\Pi$ and the global optimal policy, and the estimation bias of the feature visitation, which will be discussed in the following sections.

### 4.1 Policy Construction

In this subsection, we introduce how to construct a finite policy set $\Pi$ such that the global optimal policy $\pi^*$ can be approximated by elements in $\Pi$. The policy construction method mainly follows Appendix A.3 in Wagenmaker and Jamieson (2022) but with refined analysis for the adversarial setting.

We consider the *linear softmax* policy class. Specifically, given a parameter $w = \{w_h\}_{h=1}^H$ where $w_h \in \mathbb{R}^d$, the induced policy $\pi^w$ would select action $a \in \mathcal{A}$ at state $s$ with probability

$$\pi_h^w(a|s) = \frac{\exp\left(\eta\langle\phi(s,a), w_h\rangle\right)}{\sum_{a'}\exp\left(\eta\langle\phi(s,a'), w_h\rangle\right)} \,. \tag{4}$$

---

**Algorithm 1** GeometricHedge for Linear Adversarial MDP Policies (GLAP)

---

1: Input: policy class $\Pi$ with feature visitation estimators $\{\hat{\phi}_{\pi,h}\}_{h=1}^H$ for any $\pi \in \Pi$, confidence $\delta$, exploration parameter $\gamma \leq 1/2$

2: Initialize:   $\forall \pi \in \Pi, w_1(\pi) = 1, W_1 = |\Pi|; \eta = \gamma/dH^2$

3: **for** $h = 1, 2, \cdots, H$ **do**

4:    Compute the G-optimal design $g_h(\pi)$ on the set of feature visitations: $\{\hat{\phi}_{\pi,h}, \pi \in \Pi\}$. Denote $g(\pi) = \frac{1}{H}\sum_{h=1}^H g_h(\pi)$

5: **end for**

6: **for** each episode $k = 1, 2, \ldots, K$ **do**

7:    Compute the probabilities for any policy $\pi \in \Pi$:

$$p_k(\pi) = (1 - \gamma)\frac{w_k(\pi)}{W_k} + \gamma g(\pi)$$

8:    Select policy $\pi_k \sim p_k$ and observe losses $\ell_{k,h}(s_{k,h}, a_{k,h})$, for any $h \in [H]$

9:    Calculate the loss vector and value function estimators:

$$\hat{\theta}_{k,h} = \hat{\Sigma}_{k,h}^{-1}\hat{\phi}_{\pi_k,h}\ell_{k,h}(s_{k,h}, a_{k,h}) \,, \hat{\ell}_{k,h}^\pi = \hat{\phi}_{\pi,h}^\top\hat{\theta}_{k,h} \,, \hat{V}_k^\pi = \sum_{h=1}^H \hat{\ell}_{k,h}^\pi \,, \ \forall \pi \in \Pi$$

   where $\hat{\Sigma}_{k,h} = \sum_\pi p_k(\pi)\hat{\phi}_{\pi,h}\hat{\phi}_{\pi,h}^\top$

10:    Compute the optimistic estimate of the loss function

$$\tilde{V}_k^\pi = \sum_{h=1}^H \left(\hat{\ell}_{k,h}^\pi - 2\hat{\phi}_{\pi,h}^\top\hat{\Sigma}_{k,h}^{-1}\hat{\phi}_{\pi,h}\sqrt{\frac{H\log\left(\frac{1}{\delta}\right)}{dK}}\right) \,, \ \forall \pi \in \Pi$$

11:    Update the selecting probability using the loss estimators

$$\forall \pi \in \Pi, w_{k+1}(\pi) = w_k(\pi)\exp\left(-\eta\tilde{V}_k^\pi\right), W_{k+1} = \sum_{\pi \in \Pi}w_{k+1}(\pi)$$

12: **end for**

---

The advantage of such a policy class is that given two parameters $w$ and $u$, the value difference between policies induced by them can be upper bounded by the difference between $w$ and $u$, i.e.,

$$\left|V^{\pi^w} - V^{\pi^u}\right| \leq 2dH\eta\sum_{h=1}^H \|w_h - u_h\|_2 \,,$$

where $\eta > 0$ is the parameter in Equation 4. Recall that the structure of linear MDP guarantees that the optimal value can be represented by a linear function, i.e., there exists a weight vector $w^* = \{w_h^*\}_{h=1}^H$ where $\|w_h^*\|_2 \leq 2H\sqrt{d}$ such that $Q_h^*(s, a) = \langle\phi(s, a), w_h^*\rangle, \forall h \in [H]$ (Jin et al., 2020b). Then by constructing a parameter covering $\mathcal{W}$ over $\mathcal{B}^d(2H\sqrt{d})$ for some accuracy $\epsilon$, we can ensure that the optimal parameter $w^*$ can be approximated by a parameter $w \in \mathcal{W}^H$, i.e., $\sum_h\|w_h - w_h^*\|_2 \leq H\epsilon$. Further based on the above property, the optimal value $V^*$ can be approximated by the value $V^{\pi^w}$ of some linear softmax policy $\pi^w$ with $w \in \mathcal{W}^H$. The result of constructing such a policy covering is shown briefly in the following lemma.

**Lemma 1.** *In the linear MDP problem, there exists a finite policy class $\Pi$ with log cardinality $\log|\Pi| = \mathcal{O}\left(dH^2\log K\right)$, such that the regret $\mathrm{Reg}(K;\Pi)$ compared with the optimal policy in $\Pi$ is close to the regret $\mathrm{Reg}(K)$ compared with the global optimal policy, i.e.,*

$$\mathrm{Reg}(K) = \sum_{k=1}^K \left(V_k^{\pi_k} - V_k^{\pi^*}\right) \leq \sum_{k=1}^K V_k^{\pi_k} - \min_{\pi \in \Pi}\sum_{k=1}^K V_k^\pi + 1 =: \mathrm{Reg}(K;\Pi) + 1 \,.$$

The detailed analysis can be found in the Appendix C.

## 4.2 Feature Visitation Estimation

In this subsection, we discuss how to deal with unavailable feature visitations of policies. Our approach is to estimate the feature visitation $\{\phi_{\pi,h}\}_{h=1}^{H}$ for each policy $\pi$ and use these estimated features as input of Algorithm 1. The feature estimating process is described in Algorithm 2, which is called the feature visitation estimation oracle.

---

**Algorithm 2** FEATURE VISTATION ESTIMATION ORACLE

---

1: Input: policy set $\Pi$, tolerance $\epsilon \leq 1/2$, confidence $\delta$
2: Initialize: $\beta = 16H^2 \log(4H^2 d|\Pi|/\delta)$, $\hat{\phi}_{\pi,1} = \mathbb{E}_{a_1 \sim \pi_1(\cdot|s_1)}[\phi(s_1,a_1)]$
3: **for** $h = 1, 2, \ldots, H-1$ **do**
4:    Run procedure in Theorem 2 (Appendix B) with parameters: $\epsilon_{\exp} \leftarrow \epsilon^2/(d^3\beta)$, $\delta \leftarrow \delta/(2H)$, $\underline{\lambda} = \log(4H^2 d|\Pi|/\delta)$, $\Phi \leftarrow \Phi_h := \{\hat{\phi}_{\pi,h} : \pi \in \Pi\}$, $\gamma_\Phi = 1/(2\sqrt{d})$, and denote returned data as $\{(s_{h,\tau}, a_{h,\tau}, s_{h+1,\tau})\}_{\tau=1}^{K_h}$, for $K_h$ total number of running episodes, and covariates as

$$\Lambda_h \leftarrow \sum_{\tau=1}^{K_h} \phi(s_{h,\tau}, a_{h,\tau}) \phi(s_{h,\tau}, a_{h,\tau})^\top + 1/d \cdot I$$

5:    **for** $\pi \in \Pi$ **do**
6:

$$\hat{\phi}_{\pi,h+1} \leftarrow \left(\sum_{\tau=1}^{K_h} \phi_{\pi,h+1}(s_{h+1,\tau}) \phi_{h,\tau}^\top \Lambda_h^{-1}\right) \hat{\phi}_{\pi,h},$$

      where $\phi_{\pi,h}(s) = \mathbb{E}_{a \sim \pi_h(\cdot|s)}[\phi(s,a)]$.
7:    **end for**
8: **end for**
9: **return** $\Phi := \{\hat{\phi}_{\pi,h} : \pi \in \Pi, h = 1, 2, \cdots, H\}$

---

Under linear MDP, for any policy $\pi$, we can first decompose its feature visitation at step $h$ as

$$\phi_{\pi,h} = \mathbb{E}_\pi[\phi(s_h, a_h)] = \int \phi_{\pi,h}(s) d\mu_{h-1}^\top(s) \mathbb{E}_\pi[\phi(s_{h-1}, a_{h-1})]$$
$$= \mathcal{T}_{\pi,h} \phi_{\pi,h-1}$$
$$= \mathcal{T}_{\pi,h} \mathcal{T}_{\pi,h-1} \cdots \mathcal{T}_{\pi,2} \phi_{\pi,1},$$

where $\mathcal{T}_{\pi,h} = \int \phi_{\pi,h}(s) d\mu_{h-1}(s)$ is the transition operator and $\phi_{\pi,1} = \mathbb{E}_\pi[\phi(s_1, a_1)]$ can be directly computed based on policy $\pi$. Thus, to estimate $\phi_{\pi,h}$ for each step $h$, we need to estimate all transition operator $\{\mathcal{T}_{\pi,h}\}_{h=1}^{H}$.

We use the least square method to estimate the transition operator. Consider currently we have collected $K$ trajectories, then the estimated value is given by

$$\hat{\mathcal{T}} \in \operatorname*{argmin}_{\mathcal{T}} \sum_{\tau=1}^{K} (\mathcal{T}\phi(s_{h-1,\tau}, a_{h-1,\tau}) - \sum_a \pi(a|s_h)\phi(s_h, a))^2 + \|\mathcal{T}\|^2,$$

and the closed-form solution is that

$$\hat{\mathcal{T}}_{\pi,h} = \left(\sum_{\tau=1}^{K} \phi(s_{h-1,\tau}, a_{h-1,\tau})\phi_{\pi,h}(s_{h,\tau})\right) \Lambda_{h-1}^{-1},$$

where $\phi_{\pi,h}(s_{h,\tau}) = \sum_a \pi(a|s_h)\phi(s_h,a)$, and

$$\Lambda_{h-1} = \sum_{\tau=1}^{K} \phi(s_{h-1,\tau}, a_{h-1,\tau})\phi(s_{h-1,\tau}, a_{h-1,\tau})^\top + \lambda I \,.$$

In order to ensure the accuracy of the estimated feature visitation, we provide a guarantee for the accuracy of the estimated transition operator. The intuition is to collect enough data in each dimension of the feature space. For the design of how to collect trajectories, we adopt the reward-free technique in Wagenmaker and Jamieson (2022) and transform it to an independent feature visitation estimation oracle. At a high level, the reward free algorithm aims to collect transition data in as few interactions as possible that minimize some uncertainty measure, which is approximately $\max_\phi \|\phi\|_{\Lambda_h^{-1}}$ in our setting. To achieve this, instead of maximizing expected reward as in standard RL, the algorithm take some uncertainty objective as a surrogate reward and instead optimize over it. The detailed discussion can be found in Appendix B. Algorithm 2 satisfies the following sample complexity and accuracy guarantees.

**Lemma 2.** *For any $\epsilon > 0$, $\delta \in (0, 1)$, with probability at least $1 - \delta$, Algorithm 2 could return a feature visitation estimation for every policy $\pi \in \Pi$ in at most $\tilde{\mathcal{O}}\left(d^4 H^3/\epsilon^2\right)$ episodes that satisfies*

$$\left\|\hat{\phi}_{\pi,h} - \phi_{\pi,h}\right\|_2 \le \epsilon/\sqrt{d}, \quad \forall h \in [H]\,,$$

*where $\tilde{O}$ here hides log factors and lower order terms.*

Since the regret in an episode is less than $H$, the total regret incurred in the feature estimation process is of order $\tilde{\mathcal{O}}(d^4 H^4/\epsilon^2)$. The detailed analysis and results can be found in Appendix B.

## 5 Analysis

This section provides the regret guarantees for the proposed Algorithm 1 as well as a proof sketch. Consider Algorithm 1 with the policy set $\Pi$ constructed in Section 4.1 and the feature of policies estimated in Section 4.2 as input. Suppose we run Algorithm 1 for $K$ rounds. The regret compared with any fixed policy $\pi \in \Pi$ in these $K$ rounds can be bounded as below.

**Lemma 3.** *With probability at least $1 - \delta$, for any policy $\pi \in \Pi$,*

$$\sum_{k=1}^{K}(V_k^{\pi_k} - V_k^{\pi}) = \mathcal{O}\left(\underbrace{H\sqrt{dKH\log\frac{|\Pi|}{\delta}} + \frac{dH^2}{\gamma}\log\left(\frac{|\Pi|}{\delta}\right) + \gamma KH}_{standard\ regret\ bound\ of\ EXP3} + \underbrace{\frac{dH^2}{\gamma}\epsilon K}_{feature\ estimation\ bias}\right),$$

*where $\epsilon$ is the tolerance of the estimated feature visitation bias in Algorithm 2.*

Recall that when the policy set is constructed as Section 4.1, the difference between the global regret defined in Equation (1) and the regret compared with the best policy in $\Pi$ is just a constant. So the global regret is in the same order as the regret bound shown in Lemma 3.

Similar to previous works on linear adversarial MDP (Luo et al., 2021a;b; Sherman et al., 2023; Dai et al., 2023) that discuss the cases of whether a transition simulator is available, we define a simulator in the following assumption.

**Assumption 2** (Simulator). *The learning agent has access to a simulator such that when given a policy $\pi$, it returns a trajectory based on the MDP and policy $\pi$.*

It is worth noting that this simulator is weaker than that in Luo et al. (2021a); Dai et al. (2023) which needs to generate the next state given any state-action pair. If the learning agent has access to such a simulator described in Assumption 2, then the feature estimation process in Section 4.2 can be regarded as regret-free and the final regret is just as shown in Lemma 3. Otherwise, there is an additional $\tilde{\mathcal{O}}(d^4 H^3/\epsilon^2)$ regret term. Balancing the choice of $\gamma$ and $\epsilon$ yields the following results.

**Theorem 1.** *With the constructed policy set in Section 4.1 and the feature visitation estimation process in Section 4.2, selecting $\epsilon = \tilde{\mathcal{O}}(K^{-2/5})$, the cumulative regret satisfies*

$$\text{Reg}(K) = \mathcal{O}\left( d^{7/5} H^{12/5} K^{4/5} \log^{1/5}(K/\delta) + C_1 \right)$$

*with probability at least $1 - \delta$, where $C_1 = \text{poly}\left( d, H, \log 1/\delta, 1/\lambda^*_{\min}, \log K \right)$ is a lower order term. Additionally, if Assumption 2 holds, the regret upper bound can be improved to*

$$\text{Reg}(K) = \mathcal{O}\left( \sqrt{d^2 H^5 K \log(K/\delta)} \right) .$$

Due to the space limit, the proofs of the main results are deferred to Appendix A.

**Discussions** In this work, we adopt a new view that transforms the problem of linear adversarial MDPs into a linear optimization problem. Under this view, we propose a novel explore-exploit framework and show that it achieves the state-of-the-art result in the literature with the cost that the computational complexity depends on the size of the policy covering. Compared with previous works that focus on local properties using policy optimization, our approach provides a completely different design philosophy that globally optimizes the policy set. This framework may be of independent research interest for theoretically analyzing the linear MDP problem.

Since our work mainly aims to improve Luo et al. (2021a) with an exploratory assumption, we specifically discuss the difference between our work and theirs. It is worth noting that though both works provide guarantees for problems with exploratory assumption and transition simulators, our required assumptions are much weaker and can be implied by that in Luo et al. (2021a).

Benefiting from this new view, our work improves the regret bounds in Luo et al. (2021a) only using weaker assumptions. As shown in Table 1, our result $\tilde{\mathcal{O}}(K^{4/5})$ explicitly improve the result $\tilde{\mathcal{O}}(K^{6/7})$ in Luo et al. (2021a). And it also achieves the state-of-the-art dependence on the horizon $K$ compared with existing works (Luo et al., 2021a; Dai et al., 2023; Sherman et al., 2023). Recall that the result in Luo et al. (2021a) also depends on $\lambda$ where $\lambda$ is the minimum eigenvalue in the exploratory assumption. In real applications, for each direction in the linear space, it is reasonable that there will be a policy that visits the direction. So one could ensure the exploratory assumption by mixing these policies. However, there is no guarantee on the value of $\lambda$, and the regret depending on $1/\lambda$ can be very large when $\lambda$ is very small. Compared with Luo et al. (2021a), we remove the dependence of $\lambda$ in the main regret order term.

When a transition simulator is available, the number of calls to it (query complexity) is also important to portray the efficiency of the algorithm. Our query complexity $\mathcal{O}(K^2)$ (the detailed computation is provided in Appendix A) is much more preferred than $\mathcal{O}(KAH)^{\mathcal{O}(H)}$ in Luo et al. (2021a).

Another important advantage is that our approach can deal with the infinite action space. In some common applications including autonomous driving and robotics, the size of the action space may be infinite. However, previous works fail to apply in these settings as their approaches need to enumerate the entire action set or solve an optimization problem on the action space $\Delta(\mathcal{A})$ (Dai et al., 2023; Luo et al., 2021a; Sherman et al., 2023). We deal with this problem by putting a finite action covering over the action space when constructing the policy set in Section 4.1. More technical details can be found in Appendix C.

It is also worth noting that our result achieves the first high-probability regret bound for linear adversarial MDP, given that previous works based on policy optimization approaches only guarantee the expected regret (Luo et al., 2021a; Dai et al., 2023; Sherman et al., 2023).

To conclude, our work improves the results in Luo et al. (2021a) in terms of weaker assumptions, better regret bounds, fewer queries to the simulator (if we use one), and can deal with more general infinite action space with the cost that the computational complexity depending on the size of the policy covering. But we want to note that this dependence of computation is also standard in the reduced linear bandit problem when the cardinality of the decision set is large (Dani et al., 2007, Lemma 3.1).

# 6    Conclusion

In this paper, we investigate a new view for linear MDP that transforms the visitation feature of a policy as the feature of an arm in the linear optimization problem. Based on this view, we propose a framework that uses the estimated visitation features of policies as input and optimizes the selection probabilities of all candidate policies. With an exploratory assumption, we provide the first $\tilde{\mathcal{O}}(K^{4/5})$ regret without access to a simulator. Our algorithm enjoys a weaker assumption and a better regret bound with respect to both $K$ and $\lambda$ compared with the results in Luo et al. (2021a),. It also achieves the first high-probability type regret guarantee and can deal with the infinite action space compared with existing works for linear adversarial MDP (Luo et al., 2021a; Dai et al., 2023; Sherman et al., 2023).

Our view contributes a new approach to linear MDP, which could be of independent interest. Future implications of this technique could involve solving other adversarial settings, such as when the loss function is corrupted up to a budget, and solving robust linear MDP where the transition kernel could change over episodes. Deriving lower bounds for linear adversarial MDP is also an important direction.

## Acknowledgements

The corresponding author Shuai Li is supported by National Key Research and Development Program of China (2022ZD0114804) and National Natural Science Foundation of China (62376154, 62076161). Baoxiang Wang is partially supported by National Natural Science Foundation of China (62106213, 72150002, 72394361) and Shenzhen Science and Technology Program (RCBS20210609104356063, JCYJ20210324120011032).

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

## A    Analysis of Algorithm 1

In this section we propose the regret analysis for Algorithm 1 and prove the final regret bounds for our main algorithm. We will state the necessary concentration bounds as lemmas first and then analyze the regret, proving Lemma 3 and Theorem 1. In the following analysis, we will condition on the success of the event in Theorem 3, whose probability is at least $1 - \delta$. The following inequality will be used in the analysis and is restated in the beginning.

**Lemma 4** (Theorem 1.6 in Freedman (1975)). *Let $\mathcal{F}_0 \subset \mathcal{F}_1 \subset \cdots \subset \mathcal{F}_T$ be a filtration and let $X_1, X_2, \cdots X_T$ be random variables such that $X_t$ is $\mathcal{F}_t$ measurable, $\mathbb{E}[X_t|\mathcal{F}_{t-1}] = 0$, $|X_t| \leq b$ almost surely, and $\sum_{t=1}^{T} \mathbb{E}[X_t^2|\mathcal{F}_{t-1}] \leq V$ for some fixed $V > 0$ and $b > 0$. Then, for any $\delta \in (0,1)$, we have with probability at least $1 - \delta$,*

$$\sum_{t=1}^{T} X_t \leq 2\sqrt{V \log(1/\delta)} + b \log(1/\delta) \,.$$

To start with, we give the concentration of the feature visitation estimators returned from Algorithm 2, which will be fundamental in the following analysis. Notice that $\hat{\phi}_{\pi,1}$ can be computed directly from the initial state distribution and action distribution.

**Lemma 5.** *Suppose the estimation accuracy can be guaranteed as Theorem 3, which holds with probability at least $1 - \delta$. Then we have for all episode $k$ and steps $h$, the feature visitation estimators $\left\{\hat{\phi}_{\pi,h}, \, h = 1, 2, \cdots, H, \, \pi \in \Pi\right\}$ returned by algorithm 2 satisfy:*

$$\left|\langle \theta_{k,h}, \hat{\phi}_{\pi,h} - \phi_{\pi,h}\rangle\right| \leq \epsilon$$

*Proof.* Since $\|\theta_{k,h}\|_2 \leq \sqrt{d}$ for any $k$ and $h$, as in Definition 1, according to Theorem 3, we have:

$$\left|\langle \theta_{k,h}, \hat{\phi}_{\pi,h} - \phi_{\pi,h}\rangle\right| \leq \left\|\hat{\phi}_{\pi,h} - \phi_{\pi,h}\right\|_2 \cdot \|\theta_{k,h}\|_2 \leq \frac{\epsilon}{\sqrt{d}} \cdot \sqrt{d} \leq \epsilon \,.$$

$\square$

The following Lemma 6 and Lemma 7 will bound the magnitude for the loss and value estimators in line 9, using the properties of G-optimal design computed in line 4.

**Lemma 6.** $\left\|\hat{\phi}_{\pi,h}\right\|_{\hat{\Sigma}_{k,h}^{-1}}^2 \leq \frac{dH}{\gamma}$ *and* $\left|\hat{\phi}_{\pi,h}^{\top} \hat{\theta}_{k,h}\right| \leq \frac{dH}{\gamma}$, *for all $h$ and $\pi \in \Pi$.*

*Proof.* According to the properties of G-optimal design, we have:

$$\left\|\hat{\phi}_{\pi,h}\right\|_{\left(\sum_{\pi} g_h(\pi)\hat{\phi}_{\pi,h}\hat{\phi}_{\pi,h}^{\top}\right)^{-1}}^2 \leq d \,,$$

and $\hat{\Sigma}_{k,h} \succeq \frac{\gamma}{H} \sum_{\pi} g_h(\pi)\hat{\phi}_{\pi,h}\hat{\phi}_{\pi,h}^{\top}$. Thus we have $\left\|\hat{\phi}_{\pi,h}\right\|_{\hat{\Sigma}_{k,h}^{-1}}^2 \leq \frac{dH}{\gamma}$. So:

$$\left|\hat{\phi}_{\pi,h}^{\top} \hat{\theta}_{k,h}\right| = \left|\hat{\phi}_{\pi,h} \hat{\Sigma}_{k,h}^{-1} \hat{\phi}_{\pi_k,h} \ell_{k,h}(s_{k,h}, a_{k,h})\right| \leq \left\|\hat{\phi}_{\pi,h}\right\|_{\hat{\Sigma}_{k,h}^{-1}} \left\|\hat{\phi}_{\pi_k,h}\right\|_{\hat{\Sigma}_{k,h}^{-1}} \leq \frac{dH}{\gamma} \,, \quad \forall \pi \in \Pi \,,$$

where the second last inequality is due to the bound $\ell_{k,h}(s_h, a_h) \leq 1$. $\square$

**Lemma 7.** *With our choice of $\eta = \frac{\gamma}{2dH^2}$, when $K \geq L_0 = 4dH \log\left(\frac{|\Pi|}{\delta}\right)$, we have for all optimistic loss function estimator $\tilde{V}_k^{\pi}$, $\left|\eta \tilde{V}_k^{\pi}\right| \leq 1$.*

*Proof.* To make sure $\left|\eta \tilde{V}_k^\pi\right| \leq 1$, we notice that:

$$\left|\tilde{V}_k^\pi\right| \leq \sum_{h=1}^H \left|\hat{\phi}_{\pi,h}^\top \hat{\theta}_{k,h}\right| + \sum_{h=1}^H 2\hat{\phi}_{\pi,h}^\top \hat{\Sigma}_{k,h}^{-1} \hat{\phi}_{k,h} \sqrt{\frac{H \log 1/\delta}{dK}} . \tag{5}$$

By Lemma 6, we have

$$\left|\tilde{V}_k^\pi\right| \leq \frac{dH^2}{\gamma}\left(1 + 2\sqrt{\frac{H \log 1/\delta}{dK}}\right) .$$

When $K \geq L_0 = 4dH \log\left(\frac{|\Pi|}{\delta}\right)$, we have $\left|\tilde{V}_k^\pi\right| \leq \frac{2dH^2}{\gamma}$. Thus, our choice of $\eta = \frac{\gamma}{2dH^2}$ satisfy this constraint. $\square$

Throughout the following analysis, assuming we have run for some number of episodes $K$, we let $(\mathcal{F}_k)_{k=1}^K$ the filtration on this, with $\mathcal{F}_k$ the filtration up to and including episode $k$. Define $\mathbb{E}_k[\cdot] = \mathbb{E}[\cdot|\mathcal{F}_{k-1}]$. The next lemma will bound the bias of the loss vector estimator, thus we can bound the bias of the value function estimator.

**Lemma 8.** *Denote* $\theta_{k,h}^{\exp} = \mathbb{E}_k\left[\hat{\theta}_{k,h}\right] = \mathbb{E}_k\left[\hat{\Sigma}_{k,h}^{-1}\hat{\phi}_{\pi_k,h}\ell_{k,h}\left(s_{k,h}, a_{k,h}\right)\right]$ *as the expected value of the loss vector estimator on* $\mathcal{F}_{k-1}$. *Then we have for* $\forall \pi \in \Pi$, *with probability at least* $1 - \delta$, *the returned feature visitation estimators satisfy:*

$$\left|\langle\hat{\phi}_{\pi,h}, \theta_{k,h}^{\exp}\rangle - \langle\phi_{\pi,h}, \theta_{k,h}\rangle\right| \leq \left(\frac{dH}{\gamma} + 1\right)\epsilon \leq \frac{2dH}{\gamma}\epsilon .$$

*As a result, we also have* $\left|\langle\hat{\phi}_{\pi,h}, \theta_{k,h}^{\exp}\rangle\right| \leq \frac{2dH}{\gamma}\epsilon + 1$ .

*Proof.* Using the tower rule of expectation, we have:

$$\theta_{k,h}^{\exp} = \hat{\Sigma}_{k,h}^{-1}\mathbb{E}_k\left[\hat{\phi}_{\pi_k,h}\phi_{\pi_k,h}^\top\theta_{k,h}\right] = \hat{\Sigma}_{k,h}^{-1}\sum_{\pi'} p_k\left(\pi'\right)\hat{\phi}_{\pi',h}\phi_{\pi',h}^\top\theta_{k,h} .$$

Thus,

$$\begin{aligned}\left|\langle\hat{\phi}_{\pi,h}, \theta_{k,h}^{\exp}\rangle - \langle\hat{\phi}_{\pi,h}, \theta_{k,h}\rangle\right| &= \left|\hat{\phi}_{\pi,h}^\top\hat{\Sigma}_{k,h}^{-1}\sum_{\pi'} p_k\left(\pi'\right)\hat{\phi}_{\pi',h}\left(\phi_{\pi',h}^\top - \hat{\phi}_{\pi',h}^\top\right)\theta_{k,h}\right| \\ &\leq \sum_{\pi'} p_k\left(\pi'\right)\left|\hat{\phi}_{\pi,h}^\top\hat{\Sigma}_{k,h}^{-1}\hat{\phi}_{\pi',h}\left(\phi_{\pi',h}^\top - \hat{\phi}_{\pi',h}^\top\right)\theta_{k,h}\right| \\ &\leq \sum_{\pi'} p_k\left(\pi'\right)\frac{dH}{\gamma}\epsilon = \frac{dH}{\gamma}\epsilon ,\end{aligned}$$

where the last inequality is due to Lemma 5 and Lemma 6.
Moreover, we also have that $\left|\langle\hat{\phi}_{\pi,h} - \phi_{\pi,h}, \theta_{k,h}\rangle\right| \leq \epsilon$. Combining the two terms, we proof the lemma as:

$$\left|\langle\hat{\phi}_{\pi,h}, \theta_{k,h}^{\exp}\rangle - \langle\phi_{\pi,h}, \theta_{k,h}\rangle\right| \leq \left|\langle\hat{\phi}_{\pi,h}, \theta_{k,h}^{\exp}\rangle - \langle\hat{\phi}_{\pi,h}, \theta_{k,h}\rangle\right| + \left|\langle\hat{\phi}_{\pi,h} - \phi_{\pi,h}, \theta_{k,h}\rangle\right| \leq \left(\frac{dH}{\gamma} + 1\right)\epsilon .$$

$\square$

**Lemma 9.** *Denote* $\mathrm{DEV}_{K,\pi} = \left|\sum_{k=1}^K \hat{V}_k^\pi - V_k^\pi\right|$, *then we have with probability at least* $1 - \delta$,

$$\begin{aligned}\mathrm{DEV}_{K,\pi} \leq& 2\sum_{k=1}^K\sum_{h=1}^H \|\phi_{\pi,h}\|_{\Sigma_{k,h}^{-1}}^2 \sqrt{\frac{H \log\left(\frac{1}{\delta}\right)}{dK}} + \frac{1}{2}\sqrt{dKH \log\left(\frac{1}{\delta}\right)} \\ &+ 2\left(\frac{dH^2}{\gamma}\right)\log\left(\frac{1}{\delta}\right) + \frac{2dH^2}{\gamma}\epsilon K .\end{aligned}$$

*Proof.* First, we bound the bias of the estimated loss of each policy $\pi$ after episode $k$ in step $h$:

$$
\begin{aligned}
\hat{\ell}_{k,h}^{\pi} - \ell_{k,h}^{\pi} =& \hat{\phi}_{\pi,h}^{\top}\hat{\theta}_{k,h} - \phi_{\pi,h}^{\top}\theta_{k,h} \\
=& \left( \hat{\phi}_{\pi,h}^{\top}\hat{\theta}_{k,h} - \hat{\phi}_{\pi,h}^{\top}\theta_{k,h}^{\exp} \right) + \left( \hat{\phi}_{\pi,h}^{\top}\theta_{k,h}^{\exp} - \phi_{\pi,h}\theta_{k,h} \right) \\
\leq& \left( \hat{\phi}_{\pi,h}^{\top}\hat{\theta}_{k,h} - \hat{\phi}_{\pi,h}^{\top}\theta_{k,h}^{\exp} \right) + \frac{2dH}{\gamma}\epsilon \, .
\end{aligned}
$$

The first term is a martingale difference sequence as $\mathbb{E}_k \left[ \hat{\phi}_{\pi,h}^{\top}\hat{\theta}_{k,h} - \hat{\phi}_{\pi,h}^{\top}\theta_{k,h}^{\exp} \right] = 0$ by the definition in Lemma 8. To bound its magnitude, notice that $\left| \hat{\phi}_{\pi,h}^{\top}\hat{\theta}_{k,h} - \hat{\phi}_{\pi,h}^{\top}\theta_{k,h}^{\exp} \right| \leq \left| \hat{\phi}_{\pi,h}^{\top}\hat{\theta}_{k,h} \right| + \left| \phi_{\pi,h}^{\top}\theta_{k,h} \right| + \left| \hat{\phi}_{\pi,h}^{\top}\theta_{k,h}^{\exp} - \phi_{\pi,h}^{\top}\theta_{k,h} \right| \leq \frac{dH}{\gamma} + 1 + \frac{2dH}{\gamma}\epsilon \leq \frac{2dH}{\gamma}$ according to 5 and 6. Its variance is also bounded as:

$$
\begin{aligned}
\mathbb{E}_k \left[ \left( \hat{\phi}_{\pi,h}^{\top}\hat{\theta}_{k,h} - \hat{\phi}_{\pi,h}^{\top}\theta_{k,h}^{\exp} \right)^2 \right] \leq& \mathbb{E}_k \left[ \left( \hat{\phi}_{\pi,h}^{\top}\hat{\theta}_{k,h} \right)^2 \right] \\
=& \mathbb{E}_k \left[ \hat{\phi}_{\pi,h}^{\top}\hat{\theta}_{k,h}\hat{\theta}_{k,h}^{\top}\hat{\phi}_{\pi,h} \right] \\
=& \mathbb{E}_k \left[ \ell_{k,h} \left( s_{k,h}, a_{k,h} \right)^2 \hat{\phi}_{\pi,h}^{\top}\hat{\Sigma}_{k,h}^{-1}\hat{\phi}_{\pi_k,h}\hat{\phi}_{\pi_k,h}^{\top}\hat{\Sigma}_{k,h}^{-1}\hat{\phi}_{\pi,h} \right] \\
\leq& \left\| \hat{\phi}_{\pi,h} \right\|_{\hat{\Sigma}_{k,h}^{-1}}^2 \, ,
\end{aligned}
$$

where the last inequality is due to the fact $\mathbb{E}_k \left[ \hat{\phi}_{\pi_k,h}\hat{\phi}_{\pi_k,h}^{\top} \right] = \hat{\Sigma}_{k,h}$ and $|\ell_{k,h} \left( s_{k,h}, a_{k,h} \right)| \leq 1$. Using the Freedman inequality, we acquire with probability al least $1 - \delta$:

$$
\begin{aligned}
\text{DEV}_{K,\pi} \leq& \sum_{h=1}^{H} \left| \sum_{k=1}^{K} \hat{\ell}_{k,h}^{\pi} - \ell_{k,h}^{\pi} \right| \\
\leq& \sum_{h=1}^{H} \left[ 2\sqrt{\sum_{k=1}^{K} \left\| \hat{\phi}_{\pi,h} \right\|_{\hat{\Sigma}_{k,h}^{-1}}^2 \log \left( \frac{1}{\delta} \right)} + \frac{2dH}{\gamma}\log \left( \frac{1}{\delta} \right) + \frac{2dH}{\gamma}\epsilon K \right] \\
\leq& 2\sqrt{H \sum_{k=1}^{K}\sum_{h=1}^{H} \left\| \hat{\phi}_{\pi,h} \right\|_{\hat{\Sigma}_{k,h}^{-1}}^2 \log \left( \frac{1}{\delta} \right)} + \frac{2dH^2}{\gamma}\log \left( \frac{1}{\delta} \right) + \frac{2dH^2}{\gamma}\epsilon K \\
\leq& 2\sum_{k=1}^{K}\sum_{h=1}^{H} \left\| \hat{\phi}_{\pi,h} \right\|_{\hat{\Sigma}_{k,h}^{-1}}^2 \sqrt{\frac{H \log \left( \frac{1}{\delta} \right)}{dK}} + \frac{1}{2}\sqrt{dKH \log \left( \frac{1}{\delta} \right)} + 2\frac{dH^2}{\gamma}\log \left( \frac{1}{\delta} \right) + \frac{2dH^2}{\gamma}\epsilon K \, ,
\end{aligned}
$$

where the second last and last inequality is due to the Cauchy Schwartz inequality and GM-AM inequality.

$\square$

**Lemma 10.** *We bound the gap between the actual regret and the expected estimated regret. With probability at least $1 - \delta$,*

$$
\sum_{k=1}^{K} V_k^{\pi_k} - \sum_{k=1}^{K}\sum_{\pi} p_k \left( \pi \right) \tilde{V}_k^{\pi} \leq H \left( \sqrt{d} + \frac{2dH}{\gamma}\epsilon + 1 \right) \sqrt{2K \log \frac{1}{\delta}} + \frac{8dH^2}{3\gamma}\log \frac{1}{\delta} + 2H\sqrt{dKH \log \frac{1}{\delta}} + \frac{2dH^2}{\gamma}\epsilon K \, .
$$

*Proof.* Denote $\bar{\phi}_{k,h} = \sum_{\pi} p_k \left( \pi \right) \hat{\phi}_{\pi,h}$, we have that :

$$
\begin{aligned}
\ell_{k,h}^{\pi_k} - \sum_{\pi} p_k \left( \pi \right) \hat{\ell}_k^{\pi} =& \phi_{\pi_k,h}^{\top}\theta_{k,h} - \bar{\phi}_{k,h}^{\top}\hat{\theta}_{k,h} \\
=& \left( \phi_{\pi_k,h}^{\top}\theta_{k,h} - \hat{\phi}_{\pi_k,h}^{\top}\theta_{k,h}^{\exp} \right) + \left( \hat{\phi}_{\pi_k,h}^{\top}\theta_{k,h}^{\exp} - \bar{\phi}_{k,h}^{\top}\hat{\theta}_{k,h} \right)
\end{aligned}
$$

$$\leq \frac{2dH}{\gamma}\epsilon + \left(\hat{\phi}_{\pi_k,h}^\top \theta_{k,h}^{\exp} - \bar{\phi}_{k,h}^\top \hat{\theta}_{k,h}\right).$$

Notice that $\mathbb{E}_k\left[\hat{\phi}_{\pi_k,h}^\top \theta_{k,h}^{\exp}\right] = \mathbb{E}_k\left[\bar{\phi}_{k,h}^\top \hat{\theta}_{k,h}\right]$. We bound its conditional variance as follows:

$$\sqrt{\mathbb{E}_k\left[\left(\hat{\phi}_{\pi_k,h}^\top \theta_{k,h}^{\exp} - \bar{\phi}_{k,h}^\top \hat{\theta}_{k,h}\right)^2\right]} \leq \sqrt{\mathbb{E}_k\left[\left(\hat{\phi}_{\pi_k,h}^\top \theta_{k,h}^{\exp}\right)^2\right]} + \sqrt{\mathbb{E}_k\left[\left(\bar{\phi}_{k,h}^\top \hat{\theta}_{k,h}\right)^2\right]} \tag{6}$$

$$\leq \frac{2dH}{\gamma}\epsilon + 1 + \sqrt{\bar{\phi}_{k,h}^\top \hat{\Sigma}_{k,h}^{-1} \bar{\phi}_{k,h}} \tag{7}$$

$$\leq \frac{2dH}{\gamma}\epsilon + 1 + \sqrt{\sum_\pi p_k(\pi)\hat{\phi}_{\pi,h}^\top \hat{\Sigma}_{k,h}^{-1} \hat{\phi}_{\pi,h}} \tag{8}$$

$$= \frac{2dH}{\gamma}\epsilon + 1 + \sqrt{d}, \tag{9}$$

where inequality 6 is due to Cauchy Schwartz inequality and 8 is due to Jensen inequality. Moreover, $\left|\hat{\phi}_{\pi_k,h}^\top \theta_{k,h}^{\exp} - \bar{\phi}_{k,h}^\top \hat{\theta}_{k,h}\right| \leq \frac{2dH}{\gamma}$. Applying Bernstein's Inequality, we obtain with probability at least $1 - \delta$,

$$\sum_{k=1}^K V_k^{\pi_k} - \sum_{k=1}^K \sum_\pi p_k(\pi)\hat{V}_k^\pi \leq \sum_{h=1}^H \sum_{k=1}^K \left(\ell_{k,h}^{\pi_k} - \sum_\pi p_k(\pi)\hat{\ell}_k^\pi\right)$$

$$\leq H\left(\sqrt{d} + \frac{2dH}{\gamma}\epsilon + 1\right)\sqrt{2K\log\frac{1}{\delta}} + \frac{8}{3}\frac{dH^2}{\gamma}\log\frac{1}{\delta} + \frac{2dH^2}{\gamma}\epsilon K.$$

Since $\hat{V}_k^\pi - \tilde{V}_k^\pi = \sum_{h=1}^H 2\hat{\phi}_{\pi,h}^\top \hat{\Sigma}_{k,h}^{-1} \hat{\phi}_{k,h}\sqrt{\frac{H\log 1/\delta}{dK}}$, we have:

$$\sum_{k=1}^K \sum_\pi p_k(\pi)\left(\hat{V}_k^\pi - \tilde{V}_k^\pi\right) = \sum_{k=1}^K \sum_{h=1}^H \sum_\pi 2p_k(\pi)\hat{\phi}_{\pi,h}^\top \hat{\Sigma}_{k,h}^{-1} \hat{\phi}_{k,h}\sqrt{\frac{H\log 1/\delta}{dK}}$$

$$= 2H\sqrt{dKH\log 1/\delta}.$$

Combining the two terms, we prove this lemma. $\qquad\square$

**Lemma 11.** *With probability at least $1 - \delta$, we have:*

$$\sum_{k=1}^K \sum_\pi p_k(\pi)\left(\tilde{V}_k^\pi\right)^2 \leq 2dKH^2 + 2\frac{dH^3}{\gamma}\sqrt{2K\log\left(\frac{1}{\delta}\right)} + \frac{8dH^3\log\left(\frac{1}{\delta}\right)}{\gamma}.$$

*Proof.*

$$\sum_\pi p_k(\pi)\left(\tilde{V}_k^\pi\right)^2 \leq \sum_\pi p_k(\pi)\left[2\left(\hat{V}_k^\pi\right)^2 + 2\left(\sum_{h=1}^H 2p_k(\pi)\hat{\phi}_{\pi,h}^\top \hat{\Sigma}_{k,h}^{-1} \hat{\phi}_{k,h}\sqrt{\frac{H\log 1/\delta}{dK}}\right)^2\right]$$

$$\leq \sum_\pi p_k(\pi)\left(2\left(\hat{V}_k^\pi\right)^2 + 2H\sum_{h=1}^H 4\left\|\hat{\phi}_{\pi,h}\right\|_{\hat{\Sigma}_{k,h}^{-1}}^2 \hat{\phi}_{\pi,h}^\top \hat{\Sigma}_{k,h}^{-1} \hat{\phi}_{\pi,h}\frac{H\log\left(\frac{1}{\delta}\right)}{dK}\right)$$

$$\leq \sum_\pi p_k(\pi)\left(2\left(\hat{V}_k^\pi\right)^2 + 8\frac{dH}{\gamma}\sum_{h=1}^H \hat{\phi}_{\pi,h}^\top \hat{\Sigma}_{k,h}^{-1} \hat{\phi}_{\pi,h}\frac{H\log\left(\frac{1}{\delta}\right)}{dK}\right)$$

$$= 2\sum_\pi p_k(\pi)\left(\hat{V}_k^\pi\right)^2 + \frac{8dH^3\log 1/\delta}{\gamma K}. \tag{10}$$

Since $\left(\hat{V}_k^\pi\right)^2 \leq H \sum_{h=1}^H \left(\hat{\ell}_{k,h}^\pi\right)^2$, we bound the first term as follows.

$$\sum_\pi p_k\left(\pi\right)\left(\hat{\ell}_{k,h}^\pi\right)^2 \leq \sum_\pi p_k\left(\pi\right)\hat{\theta}_{k,h}^\top \hat{\phi}_{\pi,h}\hat{\phi}_{\pi,h}^\top \hat{\theta}_{k,h} \leq \hat{\phi}_{\pi_k,h}^\top \hat{\Sigma}_{k,h}^{-1}\hat{\phi}_{\pi_k,h}\,.$$

Its conditional expectation is $\mathbb{E}_k\left[\hat{\phi}_{\pi_k,h}^\top \hat{\Sigma}_{k,h}^{-1}\hat{\phi}_{\pi_k,h}\right] = \sum_\pi p_k\left(\pi\right)\hat{\phi}_{\pi,h}^\top \hat{\Sigma}_{k,h}^{-1}\hat{\phi}_{\pi,h} = d$, and also $\left|\hat{\phi}_{\pi_k,h}^\top \hat{\Sigma}_{k,h}^{-1}\hat{\phi}_{\pi_k,h}\right| \leq \frac{dH}{\gamma}$. Thus, applying the Hoeffding bound, we have with probability at least $1-\delta$,

$$\sum_{k=1}^K \sum_\pi p_k\left(\pi\right)\left(\hat{V}_k^\pi\right)^2 \leq H \sum_{h=1}^H dK + \frac{dH}{\gamma}\sqrt{2K\log 1/\delta} = dkH^2 + \frac{dH^3}{\gamma}\sqrt{2K\log 1/\delta}\,.$$

Plugging it into 10, we finish our proof. $\qquad\square$

*Proof of Lemma 3.* Now we are ready to start analyzing the regret. Using classical potential function analysis techniques in similar algorithms, we have:

$$\begin{aligned}
\log\left(\frac{W_{K+1}}{W_1}\right) &= \sum_{k=1}^K \log\left(\frac{W_{k+1}}{W_k}\right) \\
&= \sum_{k=1}^K \log\left(\sum_\pi \frac{w_k(\pi)}{W_k}\exp\left(-\eta\tilde{V}_k^\pi\right)\right) \\
&\leq \sum_{k=1}^K \log\left(\sum_\pi \frac{p_k(\pi)-\gamma g_\pi}{1-\gamma}\left(1-\eta\tilde{V}_k^\pi + \eta^2\left(\tilde{V}_k^\pi\right)^2\right)\right) \qquad (11) \\
&\leq \sum_{k=1}^K \sum_\pi \frac{p_k(\pi)-\gamma g_\pi}{1-\gamma}\left(-\eta\tilde{V}_k^\pi + \eta^2\left(\tilde{V}_k^\pi\right)^2\right) \\
&\leq \frac{\eta}{1-\gamma}\left[\sum_{k=1}^K \sum_\pi -p_k(\pi)\tilde{V}_k^\pi + \gamma\sum_{k=1}^K \sum_\pi g(\pi)\tilde{V}_k^\pi + \eta\sum_{k=1}^K \sum_\pi p_k(\pi)\left(\tilde{V}_k^\pi\right)^2\right]\,, \qquad (12)
\end{aligned}$$

where inequality 11 is from $\left|\eta\tilde{V}_k^\pi\right| \leq 1$ guaranteed by Lemma 7. Using Lemma 9, we can bound the second term as:

$$\begin{aligned}
\sum_{k=1}^K \tilde{V}_k^\pi &\leq \sum_{k=1}^K V_k^\pi + \text{DEV}_{k,\pi} - \sum_{k=1}^K \sum_{h=1}^H 2\hat{\phi}_{\pi,h}^\top \hat{\Sigma}_{k,h}^{-1}\hat{\phi}_{\pi,h}\sqrt{H\frac{\log\left(\frac{1}{\delta}\right)}{dK}} \\
&\leq \sum_{k=1}^K V_k^\pi + \frac{1}{2}\sqrt{dKH\log\left(\frac{1}{\delta}\right)} + 2\left(\frac{dH^2}{\gamma}\right)\log\left(\frac{1}{\delta}\right) + \frac{2dH^2}{\gamma}\epsilon K \qquad (13) \\
&\leq KH + \frac{1}{2}\sqrt{dKH\log\left(\frac{1}{\delta}\right)} + 2\left(\frac{dH^2}{\gamma}\right)\log\left(\frac{1}{\delta}\right) + \frac{2dH^2}{\gamma}\epsilon K\,.
\end{aligned}$$

Plugging Lemma 10, Lemma 11 and Equation (13) into Equation (12), notice we condition on $\gamma \leq 1/2$, we obtain:

$$\begin{aligned}
\log\left(\frac{W_{K+1}}{W_1}\right) &\leq -\eta\sum_{k=1}^K V_k^{\pi_k} + 2\eta^2\left[2dKH^2 + 2\frac{dH^3}{\gamma}\sqrt{2K\log\left(\frac{1}{\delta}\right)} + \frac{8dH^3\log\left(\frac{1}{\delta}\right)}{\gamma}\right] \\
&\quad + 2\eta\gamma KH + \eta\left[\frac{1}{2}\sqrt{dKH\log\left(\frac{1}{\delta}\right)} + 2\left(\frac{dH^2}{\gamma}\right)\log\left(\frac{1}{\delta}\right) + \frac{2dH^2}{\gamma}\epsilon K\right] \qquad (14) \\
&\quad + 2\eta\left[H\left(\sqrt{d} + \frac{2dH}{\gamma}\epsilon + 1\right)\sqrt{2K\log\frac{1}{\delta}} + \frac{8}{3}\frac{dH^2}{\gamma}\log\frac{1}{\delta} + 2H\sqrt{dKH\log\frac{1}{\delta}} + \frac{2dH^2}{\gamma}\epsilon K\right]\,.
\end{aligned}$$

Combining terms, we have:

$$\frac{\log\left(\frac{W_{K+1}}{W_1}\right)}{\eta} \leq -\sum_{k=1}^{K} V_k^{\pi_k} + \mathcal{O}\left(H\sqrt{dKH\log\frac{1}{\delta}}\right) + \mathcal{O}\left(\frac{dH^2}{\gamma}\log\left(\frac{1}{\delta}\right) + \gamma KH\right)$$
$$+ \mathcal{O}\left(\eta dKH^2 + \frac{\eta dH^3}{\gamma}\sqrt{2K\log\left(\frac{1}{\delta}\right)} + \frac{8\eta dH^3\log\left(\frac{1}{\delta}\right)}{\gamma}\right) + \mathcal{O}\left(\frac{dH^2}{\gamma}\epsilon K\right). \tag{15}$$

Plugging $\eta = \frac{\gamma}{dH^2}$ into Equation (15), we have:

$$\frac{\log\left(\frac{W_{K+1}}{W_1}\right)}{\eta} \leq -\sum_{k=1}^{K} V_k^{\pi_k} + \mathcal{O}\left(H\sqrt{dKH\log\frac{1}{\delta}}\right) + \mathcal{O}\left(\frac{dH^2}{\gamma}\log\left(\frac{1}{\delta}\right) + \gamma KH\right) + \mathcal{O}\left(\frac{dH^2}{\gamma}\epsilon K\right). \tag{16}$$

On the other hand, we have:

$$\log\left(\frac{W_{K+1}}{W_1}\right) \geq \eta\left(\sum_{k=1}^{K} -\tilde{V}_k^{\pi}\right) - \log\left(|\Pi|\right)$$
$$\geq \eta\left(\sum_{k=1}^{K} -V_k^{\pi} - \text{DEV}_{K,\pi} + \sum_{k=1}^{K}\sum_{h=1}^{H} 2\hat{\phi}_{\pi,h}^\top \hat{\Sigma}_{k,h}^{-1}\hat{\phi}_{\pi,h}\sqrt{H\frac{\log\left(\frac{1}{\delta}\right)}{dK}}\right) - \log\left(|\Pi|\right) \tag{17}$$
$$\geq \eta\left(\sum_{k=1}^{K} -V_k^{\pi} - \frac{1}{2}\sqrt{dKH\log\left(\frac{1}{\delta}\right)} - 2\left(\frac{dH^2}{\gamma}\right)\log\left(\frac{1}{\delta}\right) - \frac{2dH^2}{\gamma}\epsilon K\right) - \log\left(|\Pi|\right).$$

Combining (16) and (17), we have:

$$\sum_{k=1}^{K} V_k^{\pi_k} - V_k^{\pi} \leq \mathcal{O}\left(H\sqrt{dKH\log\frac{1}{\delta}} + \frac{dH^2}{\gamma}\log\left(\frac{1}{\delta}\right) + \gamma KH\right) + \mathcal{O}\left(\frac{dH^2}{\gamma}\epsilon K\right) + \frac{\log\left(|\Pi|\right)}{\eta}. \tag{18}$$

Choosing $\eta = \frac{\gamma}{dH^2}$ and combining terms, we obtain for any policy $\pi \in \Pi$, with probability at least $1 - \delta$:

$$\sum_{k=1}^{K} V_k^{\pi_k} - V_k^{\pi} = \mathcal{O}\left(H\sqrt{dKH\log\frac{|\Pi|}{\delta}} + \frac{dH^2}{\gamma}\log\left(\frac{|\Pi|}{\delta}\right) + \gamma KH\right) + \mathcal{O}\left(\frac{dH^2}{\gamma}\epsilon K\right). \tag{19}$$

$\square$

We will then present the proof of Theorem 1 based on Equation (19). Notice we condition on $K$ being large enough so that the optimal parameters $\gamma$ and $\epsilon$ set below are smaller than $\frac{1}{2}$, satisfying the requirements of the algorithm, while the cases of $K$ being small is trivial.

- In the case when we have access to a simulator, the total regret occurred while we execute the policies in $\Pi$. Set the parameters as $\epsilon \leftarrow dH^2\log\frac{K}{\delta}/K$, $\gamma \leftarrow \sqrt{dH\log\left(\frac{|\Pi|}{\delta}\right)}/K$ and using the properties of $\Pi$ in Lemma 18, the total regret is bounded as:

$$\text{Reg}(K) \leq \text{Reg}\left(K;\Pi\right) + 1 = \max_{\pi\in\Pi}\left(\sum_{k=1}^{K} V_k^{\pi_k} - V_k^{\pi}\right) + 1 = \mathcal{O}\left(\sqrt{d^2 H^5 K\log\frac{K}{\delta}}\right).$$

Also, according to corollary 1, the total number of episodes run on the simulator is in the order of $\tilde{\mathcal{O}}\left(d^3 HK^2\right)$.

- When we don't have access to a simulator, we have to take account of the regret occurred while we estimate the feature visitation of each policy. According to corollary 1, the additional regret is in the order of $\mathcal{O}\left(\frac{d^4 H^4}{\epsilon^2}\log\frac{H^2 d|\Pi|}{\delta} + C_1\right)$. By our construction of policy set $\Pi$ in Lemma 18, the total regret is bounded as:

$$\text{Reg}(K) = \mathcal{O}\left(\frac{d^5 H^6}{\epsilon^2}\log\frac{K}{\delta} + \frac{d^2 H^4}{\gamma}\log\frac{K}{\delta} + \gamma K H + \frac{dH^2}{\gamma}\epsilon K + \sqrt{d^2 H^5 K \log\frac{K}{\delta}} + C_1\right), \quad (20)$$

with $C_1 = \text{poly}\left(d, H, \log 1/\delta, \frac{1}{\lambda_{min}^*}, \log|\Pi|, \log 1/\epsilon\right)$. Set the parameters as $\epsilon \leftarrow K^{-2/5}d^{9/5}H^{9/5}\log^{2/5}\frac{K}{\delta}$, $\gamma \leftarrow K^{-1/5}d^{7/5}H^{7/5}\log^{1/5}\frac{K}{\delta}$, the total regret is in the order of:

$$\text{Reg}(K) = \tilde{\mathcal{O}}\left(d^{7/5}H^{12/5}K^{4/5}\log^{1/5}\frac{K}{\delta}\right).$$

## B  Construct the Policy Visitation Estimators

In this section, we will propose the analysis of Algorithm 2. We then prove theorem 3 and corollary 1 as our main results, which will provide the concentration of the estimators $\hat{\phi}_{\pi,h}$ and bound the sample complexity. These results will then be used to proof the final regret bounds in Appendix A.

First, we propose the performance guarantee of the data collecting oracle, which comes directly from theorem 9 in Wagenmaker and Jamieson (2022). Denote:

$$\mathbf{XY}_{\text{opt}}(\mathbf{\Lambda}) = \max_{\phi\in\Phi}\|\phi\|_{\mathbf{A}(\mathbf{\Lambda})^{-1}}^2 \quad \text{for } \mathbf{A}(\mathbf{\Lambda}) = \mathbf{\Lambda} + \mathbf{\Lambda}_0,$$

for $\mathbf{\Lambda}_0$ be some fixed regularizer. We consider it's smooth approximation:

$$\widetilde{\mathbf{XY}}_{\text{opt}}(\mathbf{\Lambda}) = \frac{1}{\eta}\log\left(\sum_{\phi\in\Phi}e^{\eta\|\phi\|_{\mathbf{A}(\mathbf{\Lambda})^{-1}}^2}\right).$$

We also define $\mathbf{\Omega}_h := \{\mathbb{E}_{\pi\sim\omega}[\mathbf{\Lambda}_{\pi,\mathbf{h}}] : \omega\in\Delta_\pi\}$, where $\Delta_\pi$ is the set of all the distributions over all valid Markovian policies. $\mathbf{\Omega}_h$ is, then, the set of all covariance matrices realizable by distributions over policies at step $h$. Then we have

**Theorem 2.** *Considering running Algorithm 6 in* Wagenmaker and Jamieson (2022) *with some $\epsilon > 0$ and functions*

$$f_i(\mathbf{\Lambda}) \leftarrow \widetilde{\mathbf{XY}}_{opt}(\mathbf{\Lambda})$$

*for $\mathbf{\Lambda}_0 \leftarrow (T_i K_i)^{-1}\Sigma_i =: \mathbf{\Lambda}_i$ and*

$$\eta_i = \frac{2}{\gamma_\Phi}\cdot\left(1 + \|\mathbf{\Lambda}_i\|_{\text{op}}\right)\cdot\log|\Phi|$$

$$L_i = \|\mathbf{\Lambda}_i^{-1}\|_{\text{op}}^2, \quad \beta_i = 2\|\mathbf{\Lambda}_i^{-1}\|_{\text{op}}^3\left(1 + \eta_i\|\mathbf{\Lambda}_i^{-1}\|_{\text{op}}\right), \quad M_i = \|\mathbf{\Lambda}_i^{-1}\|_{\text{op}}^2$$

*where $\Sigma_i$ is the matrix returned by running Algorithm 7 in* Wagenmaker and Jamieson (2022) *with $N \leftarrow T_i K_i$, $\delta \leftarrow \delta/(2i^2)$, and some $\underline{\lambda} \geq 0$. Then with probability $1 - 2\delta$, this procedure will collect at most*

$$20\cdot\frac{\inf_{\mathbf{\Lambda}\in\Omega}\max_{\phi\in\Phi}\|\phi\|_{\mathbf{A}(\mathbf{\Lambda})^{-1}}^2}{\epsilon_{\text{exp}}} + \text{poly}\left(d, H, \log 1/\delta, \frac{1}{\lambda_{\min}^*}, \frac{1}{\gamma_\Phi}, \underline{\lambda}, \log|\Phi|, \log\frac{1}{\epsilon_{\text{exp}}}\right)$$

*episodes, where*

$$\mathbf{A}(\mathbf{\Lambda}) = \mathbf{\Lambda} + \min\left\{\frac{(\lambda_{\min}^*)^2}{d}, \frac{\lambda_{\min}^*}{d^3 H^3\log^{7/2}1/\delta}\right\}\cdot\text{poly}\log\left(\frac{1}{\lambda_{\min}^*}, d, H, \underline{\lambda}, \log\frac{1}{\delta}\right)\cdot I,$$

*and will produce covariates $\widehat{\Sigma} + \Sigma_i$ such that*

$$\max_{\phi \in \Phi} \|\phi\|^2_{\left(\widehat{\Sigma} + \Sigma_i\right)^{-1}} \leq \epsilon_{\exp}$$

*and*

$$\lambda_{\min}\left(\widehat{\Sigma} + \Sigma_i\right) \geq \max\left\{d \log 1/\delta, \; \lambda\right\} .$$

Next, we will propose the concentration analysis of our estimators and bound the total number of episodes run. Throughout this section, assuming we have run for some number of episodes K, we let $(\mathcal{F}_\tau)_{\tau=1}^K$ the filtration on this, with $\mathcal{F}_\tau$ the filtration up to and including episode $\tau$. We also let $\mathcal{F}_{\tau,h}$ denote the filtration on all episodes $\tau' < \tau$, and on steps $h' = 1, \cdots, h$ of episode $\tau$. Define

$$\phi_{\pi,h} = \mathbb{E}_\pi\left[\phi\left(s_h, a_h\right)\right], \quad \phi_{\pi,h}(s) = \sum_{a \in \mathcal{A}} \phi(s, a) \pi_h(a|s)$$

and

$$\mathcal{T} := \int \phi_{\pi,h}(s) \; d\mu_{h-1}(s)^\top .$$

We have from lemma A.7 in Wagenmaker and Jamieson (2022): $\phi_{\pi,h} = \mathcal{T}_{\pi,h} \phi_{\pi,h-1} = \cdots = \mathcal{T}_{\pi,h} \cdots \mathcal{T}_{\pi,1} \phi_{\pi,0}$. We also denote $\gamma_\Phi := \max_{\phi \in \Phi} \|\phi\|_2$.

The following Lemma 12 comes straight from lemma B.1, lemma B.2 and lemma B.3 in Wagenmaker and Jamieson (2022) and provides us with the basic concentration properties of the estimators constructed in line 6 of Algorithm 2.

**Lemma 12.** *Assume that we have collected some data $\{(s_{h-1,\tau}, a_{h-1,\tau}, s_{h,\tau})\}_{\tau=1}^K$ where, for each $\tau'$, $s_{h,\tau'}|\mathcal{F}_{h-1,\tau'}$ is independent of $\{(s_{h-1,\tau}, a_{h-1,\tau}, s_{h,\tau})\}_{\tau \neq \tau'}$. Denote $\phi_{h-1,\tau} = \phi(s_{h-1,\tau}, a_{h-1,\tau})$ and $\Lambda_{h-1} = \sum_{\tau=1}^K \phi_{h-1,\tau} \phi_{h-1,\tau}^\top + \lambda I$. Fix $\pi$ and let*

$$\hat{\mathcal{T}}_{\pi,h} = \left(\sum_{\tau=1}^K \phi_{\pi,h}(s_{h,\tau}) \phi_{h-1,\tau}^\top\right) \Lambda_{h-1}^{-1}$$

$$\hat{\phi}_{\pi,h} = \hat{\mathcal{T}}_{\pi,h} \hat{\mathcal{T}}_{\pi,h-1} \cdots \hat{\mathcal{T}}_{\pi,2} \hat{\mathcal{T}}_{\pi,1} \phi_{\pi,0} .$$

*Fix $\boldsymbol{u} \in \mathcal{S}^{d-1}$. Then with probability at least $1 - \delta$:*

$$\left|\langle \boldsymbol{u}, \phi_{\pi,h} - \hat{\phi}_{\pi,h}\rangle\right| \leq \sum_{i=1}^{h-1} \left(2\sqrt{\log \frac{2H}{\delta}} + \frac{\log \frac{2H}{\delta}}{\sqrt{\lambda_{\min}(\Lambda_i)}} + \sqrt{d\lambda}\right) \cdot \left\|\hat{\phi}_{\pi,i}\right\|_{\Lambda_i^{-1}} .$$

*Thus, with probability at least $1 - \delta$,*

$$\left\|\hat{\phi}_{\pi,h} - \phi_{\pi,h}\right\|_2 \leq d\sum_{h'=1}^{h-1} \left(2\sqrt{\log \frac{2Hd}{\delta}} + \frac{\log \frac{2Hd}{\delta}}{\sqrt{\lambda_{\min}(\Lambda_{h'})}} + \sqrt{d\lambda}\right) \cdot \left\|\hat{\phi}_{\pi,h'}\right\|_{\Lambda_{h'}^{-1}}$$

**Lemma 13.** *Let $\varepsilon_{\text{est}}^h$ denote the event on which, for all $\pi \in \Pi$, the feature visitation estimates returned by line 6 satisfy:*

$$\left\|\hat{\phi}_{\pi,h+1} - \phi_{\pi,h+1}\right\|_2 \leq d\sum_{h'=1}^{h-1} \left(3\sqrt{\log \frac{4H^2 d|\Pi|}{\delta}} + \frac{\log \frac{4H^2 d|\Pi|}{\delta}}{\sqrt{\lambda_{\min}(\Lambda_{h'})}}\right) \cdot \left\|\hat{\phi}_{\pi,h'}\right\|_{\Lambda_{h'}^{-1}}$$

*Then $\mathbb{P}\left[\left(\varepsilon_{\text{est}}^h\right)^c\right] \leq \frac{\delta}{2H}$ .*

*Proof.* Following similar analysis in lemma B.5 in Wagenmaker and Jamieson (2022), we also have the data collected in Theorem 2 satisfy the independent requirements of Lemma 12. The result follows by setting $\lambda = 1/d$ in Lemma 12. $\square$

**Lemma 14.** *Let $\varepsilon_{\exp}^h$ denote the event on which: The total number of episodes run in line 4 is at most*

$$C\frac{d^3\inf_{\mathbf{\Lambda}\in\Omega_h}\max_{\phi\in\Phi_h}\|\phi\|^2_{\mathbf{A}(\mathbf{\Lambda})^{-1}}}{\epsilon^2/\beta} + \mathrm{poly}\left(d, H, \log 1/\delta, \frac{1}{\lambda_{\min}^*}, \log|\Pi|, \log 1/\epsilon\right)$$

*episodes. The covariates returned by line 4, $\mathbf{\Lambda}_h$, satisfy:*

$$\max_{\phi\in\Phi_h}\|\phi\|^2_{\mathbf{\Lambda}_h^{-1}} \le \frac{\epsilon^2}{d^3\beta}, \quad \lambda_{\min}\left(\mathbf{\Lambda}_h\right) \ge \log\frac{4H^2d|\Pi|}{\delta}. \tag{21}$$

*Then $\mathbb{P}\left[\left(\varepsilon_{\exp}^h\right)^c \cap \varepsilon_{\mathrm{est}}^{h-1} \cap \left(\cap_{i=1}^{h-1}\varepsilon_{\exp}^i\right)\right] \le \frac{\delta}{2H}$.*

*Proof.* By Lemma 15, on the event $\varepsilon_{\mathrm{est}}^{h-1}\cap\left(\cap_{i=1}^{h-1}\varepsilon_{\exp}^i\right)$ we can bound $\left\|\hat{\phi}_{\pi,h+1} - \phi_{\pi,h+1}\right\|_2 \le \epsilon/\sqrt{d}$. Remember that we condition on $K$ being large enough so that we have $\epsilon \le 1/2$. Also, we can lower bound $\|\phi_{\pi,h}\|_2 \ge 1/\sqrt{d}$ from lemma A.6 in Wagenmaker and Jamieson (2022). Thus,

$$\left\|\hat{\phi}_{\pi,h}\right\|_2 \ge \|\phi_{\pi,h}\|_2 - \left\|\hat{\phi}_{\pi,h+1} - \phi_{\pi,h+1}\right\|_2 \ge 1/\sqrt{d} - \epsilon/\sqrt{d} \ge 1/\left(2\sqrt{d}\right).$$

So the choice of $\gamma_\Phi = \frac{1}{2\sqrt{d}}$ is valid. The result then follows by applying Theorem 2 with our chosen parameters. $\square$

**Lemma 15.** *On the event $\varepsilon_{\mathrm{est}}^h \cap \left(\cap_{i=1}^h\varepsilon_{\exp}^i\right)$, for all $\pi\in\Pi$,*

$$\left\|\hat{\phi}_{\pi,h+1} - \phi_{\pi,h+1}\right\|_2 \le \epsilon/\sqrt{d}.$$

*Proof.* On $\varepsilon_{exp}^i$, we can bound:

$$\lambda_{\min}\left(\mathbf{\Lambda}_i\right) \ge \log\frac{4H^2d|\Pi|}{\delta},$$

$$\left\|\hat{\phi}_{\pi,i}\right\|_{\mathbf{\Lambda}_i^{-1}} \le \frac{\epsilon}{d\sqrt{d\beta}} = \frac{\epsilon}{4Hd\sqrt{d\log\frac{4H^2d|\Pi|}{\delta}}}.$$

so that:

$$\begin{aligned}
\left\|\hat{\phi}_{\pi,h+1} - \phi_{\pi,h+1}\right\|_2 &\le d\sum_{h'=1}^{h-1}\left(3\sqrt{\log\frac{4H^2d|\Pi|}{\delta}} + \frac{\log\frac{4H^2d|\Pi|}{\delta}}{\sqrt{\lambda_{\min}\left(\mathbf{\Lambda}_{h'}\right)}}\right)\cdot\left\|\hat{\phi}_{\pi,h'}\right\|_{\mathbf{\Lambda}_{h'}^{-1}} \\
&\le d\sum_{i=1}^h 4\sqrt{\log\frac{4H^2d|\Pi|}{\delta}}\cdot\frac{\epsilon}{4Hd\sqrt{d\log\frac{4H^2d|\Pi|}{\delta}}} \\
&\le dH\cdot\frac{\epsilon}{dH\sqrt{d}} = \epsilon/\sqrt{d}.
\end{aligned}$$

$\square$

**Lemma 16.** *Define $\varepsilon_{\exp} = \cap_h\varepsilon_{\exp}^h$ and $\varepsilon_{\mathrm{est}} = \cap_h\varepsilon_{\mathrm{est}}^h$. Then $\mathbb{P}\left[\varepsilon_{est}\cap\varepsilon_{exp}\right] \ge 1-\delta$, and conditioning on $\varepsilon_{est}\cap\varepsilon_{exp}$, for all $h=1,2,\cdots,H-1$ and $\pi\in\Pi$, we have:*

$$\left\|\hat{\phi}_{\pi,h+1} - \phi_{\pi,h+1}\right\|_2 \le \epsilon/\sqrt{d}.$$

*Proof.* Obviously,

$$\varepsilon_{\exp}^c\cup\varepsilon_{\mathrm{est}}^c = \bigcup_{h=1}^{H-1}\left(\left(\varepsilon_{\exp}^h\right)^c\cup\left(\varepsilon_{\mathrm{est}}^h\right)^c\right)$$

$$= \bigcup_{h=1}^{H-1} \left(\varepsilon_{\exp}^h\right)^c \setminus \left(\left(\varepsilon_{\text{est}}^{h-1}\right)^c \cup \left(\cup_{i=1}^{h-1} \left(\varepsilon_{\exp}^i\right)^c\right)\right) \cup \bigcup_{h=1}^{H} \left(\varepsilon_{\text{est}}^h\right)^c$$

$$= \bigcup_{h=1}^{H-1} \left(\varepsilon_{\exp}^h\right)^c \cap \left(\varepsilon_{\text{est}}^{h-1} \cap \left(\cap_{i=1}^{h-1} \varepsilon_{\exp}^i\right)\right) \cup \bigcup_{h=1}^{H} \left(\varepsilon_{\text{est}}^h\right)^c .$$

Using Lemma 13 and Lemma 14, we can bound

$$\mathbb{P}\left[\varepsilon_{\text{est}}^c \cup \varepsilon_{\exp}^c\right] \le \sum_{h=1}^{H-1} \left(\mathbb{P}\left[\left(\varepsilon_{\exp}^h\right)^c \cap \varepsilon_{\text{est}}^{h-1} \cap \left(\cap_{i=1}^{h-1} \varepsilon_{\exp}^i\right)\right] + \mathbb{P}\left[\left(\varepsilon_{\text{est}}^h\right)^c\right]\right)$$

$$\le \sum_{h=1}^{H-1} 2 \cdot \frac{\delta}{2H}$$

$$\le \delta .$$

And the inequality follows by Lemma 15. $\qquad\square$

**Theorem 3.** *(Full version of Lemma 2) With probability at least $1 - \delta$, Algorithm 2 will run at most*

$$CH^2 d^3 \sum_{h=1}^{H-1} \frac{\inf_{\mathbf{\Lambda}\in\Omega_h} \max_{\pi\in\Pi} \|\phi_{\pi,h}\|_{\mathbf{\Lambda}^{-1}}^2}{\epsilon^2} \log \frac{H^2 d|\Pi|}{\delta} + \text{poly}\left(d, H, \log 1/\delta, \frac{1}{\lambda_{\min}^*}, \log |\Pi|, \log 1/\epsilon\right)$$

*episodes, and will output policy visitation estimators $\Phi = \left\{\hat{\phi}_{\pi,h} : h = 1, 2, \cdots, H, \pi \in \Pi\right\}$ with bias bounded as:*

$$\left\|\hat{\phi}_{\pi,h} - \phi_{\pi,h}\right\|_2 \le \epsilon/\sqrt{d} .$$

*Proof.* According to Lemma 16, we can condition on the event $\varepsilon_{est} \cap \varepsilon_{exp}$, thus we obtain the accuracy desired. According to Lemma 14, we have total episodes be bounded as :

$$\sum_{h=1}^{H-1} C \frac{d^3 \inf_{\mathbf{\Lambda}\in\Omega_h} \max_{\phi\in\Phi_h} \|\phi\|_{\mathbf{A}(\mathbf{\Lambda})^{-1}}^2}{\epsilon^2/\beta} + \text{poly}\left(d, H, \log 1/\delta, \frac{1}{\lambda_{\min}^*}, \log |\Pi|, \log 1/\epsilon\right)$$

$$\le \sum_{h=1}^{H-1} C \frac{d^3 \inf_{\mathbf{\Lambda}\in\Omega_h} \max_{\phi\in\Phi_h} \|\phi\|_{\mathbf{A}(\mathbf{\Lambda})^{-1}}^2}{\epsilon^2} H^2 \log \frac{H^2 d|\Pi|}{\delta} + \text{poly}\left(d, H, \log 1/\delta, \frac{1}{\lambda_{\min}^*}, \log |\Pi|, \log 1/\epsilon\right) .$$

Conditioning on $\varepsilon_{est} \cap \varepsilon_{exp}$, we have for all $\pi \in \Pi$, $\left\|\hat{\phi}_{\pi,h} - \phi_{\pi,h}\right\|_2 \le \epsilon/\sqrt{d}$, thus we can upper bound:

$$\inf_{\mathbf{\Lambda}\in\Omega_h} \max_{\phi\in\Phi_h} \|\phi\|_{\mathbf{A}(\mathbf{\Lambda})^{-1}}^2 = \inf_{\mathbf{\Lambda}\in\Omega_h} \max_{\pi\in\Pi} \left\|\hat{\phi}_{\pi,h}\right\|_{\mathbf{A}(\mathbf{\Lambda})^{-1}}^2$$

$$\le \inf_{\mathbf{\Lambda}\in\Omega_h} \max_{\pi\in\Pi} \left(2 \|\phi_{\pi,h}\|_{\mathbf{A}(\mathbf{\Lambda})^{-1}}^2 + 2 \left\|\hat{\phi}_{\pi,h} - \phi_{\pi,h}\right\|_{\mathbf{A}(\mathbf{\Lambda})^{-1}}^2\right)$$

$$\le \inf_{\mathbf{\Lambda}\in\Omega_h} \max_{\pi\in\Pi} \left(2 \|\phi_{\pi,h}\|_{\mathbf{A}(\mathbf{\Lambda})^{-1}}^2 + \frac{2\epsilon^2}{d\lambda_{\min}(\mathbf{A}(\mathbf{\Lambda}))}\right)$$

$$\le \inf_{\mathbf{\Lambda}\in\Omega_h} \max_{\pi\in\Pi} 4 \|\phi_{\pi,h}\|_{\mathbf{\Lambda}^{-1}}^2 + \frac{4\epsilon^2}{d\lambda_{\min}^*}$$

Thus, the total number of episodes is bounded as:

$$CH^2 d^3 \sum_{h=1}^{H-1} \frac{\inf_{\mathbf{\Lambda}\in\Omega_h} \max_{\pi\in\Pi} \|\phi_{\pi,h}\|_{\mathbf{\Lambda}^{-1}}^2}{\epsilon^2} \log \frac{H^2 d|\Pi|}{\delta} + \text{poly}\left(d, H, \log 1/\delta, \frac{1}{\lambda_{\min}^*}, \log |\Pi|, \log 1/\epsilon\right) . \qquad (22)$$

$\qquad\square$

From lemma B.10 in Wagenmaker and Jamieson (2022), we can bound:

$$\inf_{\mathbf{\Lambda} \in \Omega_h} \max_{\pi \in \Pi} \|\phi_{\pi,h}\|^2_{\mathbf{\Lambda}^{-1}} \le d \,.$$

Thus we have:

**Corollary 1.** *The sample complexity in Algorithm* 2 *is bounded by:*

$$\mathcal{O}\left( \frac{d^4 H^3}{\epsilon^2} \log \frac{H^2 d |\Pi|}{\delta} + C_1 \right) \,,$$

*where* $C_1 = \mathrm{poly}\left( d, H, \log 1/\delta, \frac{1}{\lambda^*_{\min}}, \log |\Pi|, \log 1/\epsilon \right)$.

## C  Construct the Policy Set

In this section we provide the proof for the policy set $\Pi$ we constructed. The construction techniques follows directly from Appendix A.3 in Wagenmaker and Jamieson (2022) and we will prove such construction also works in MDP with adversarial rewards. Our main result is stated in Lemma 18.

**Lemma 17.** *In the adversarial MDP setting, where the loss function changes in each round, the best policy of the MDP* $\mathcal{M}(\mathcal{S}, \mathcal{A}, H, \{P_h\}^H_{h=1}, \{\ell_k\}^K_{k=1})$ *in rounds* 1 *to* $K$ *from the set of all stationary policies, is the optimal policy of the MDP with a **fixed** loss function being the average. Denote the average MDP as* $\mathring{\mathcal{M}}(\mathcal{S}, \mathcal{A}, H, \{P_h\}^H_{h=1}, \mathring{\ell})$*, with the same transition kernel and the average loss* $\mathring{\ell} = \frac{1}{K} \sum^K_{k=1} \ell_k$*. That is:*

$$\text{if} \quad \pi^* = \underset{\pi}{\operatorname{argmin}} \sum_{k=1}^K V_k^\pi \,,$$

$$\text{then} \quad \pi^* = \underset{\pi}{\operatorname{argmin}} \mathring{V}^\pi \,.$$

*Where* $\mathring{V}$ *is the value function associated with the new MDP* $\mathring{\mathcal{M}}$.

*Proof.* Let $\tau_\pi = ((s_1, a_1), (s_2, a_2), \cdots (s_H, a_H))$ be the trajectory generated by following policy $\pi$ through the MDP. Denote the occupancy measure $\mu_h^\pi(s_h, a_h)$ as the probability of visiting state-action pair $(s_h, a_h)$ under trajectory $\tau_\pi$, and $\mu^\pi = (\mu_1^\pi, \mu_2^\pi, \cdots \mu_H^\pi)$.
For any stationary policy $\pi$, we have:

$$V^\pi = \sum_{h=1}^H \sum_{(s_h, a_h) \in \mathcal{S}_h \times \mathcal{A}_h} \mu_h^\pi(s_h, a_h) \ell_k(s_h, a_h) \,.$$

Since the two MDP share the same transition kernel, the occupancy measure generated by the same policy stays unchanged. So we have:

$$\begin{aligned}
\sum_{k=1}^K V_k^\pi &= \sum_{k=1}^K \sum_{h=1}^H \sum_{(s_h, a_h) \in \mathcal{S}_h \times \mathcal{A}_h} \mu_h^\pi(s_h, a_h) \ell_k(s_h, a_h) \\
&= \sum_{h=1}^H \sum_{(s_h, a_h) \in \mathcal{S}_h \times \mathcal{A}_h} \mu_h^\pi(s_h, a_h) \left( \sum_{k=1}^K l_{k,h}(s_h, a_h) \right) \\
&= \sum_{h=1}^H \sum_{(s_h, a_h) \in \mathcal{S}_h \times \mathcal{A}_h} \mu_h^\pi(s_h, a_h) \cdot K \mathring{l}(s_h, a_h) \\
&= K \mathring{V}^\pi
\end{aligned}$$

So $\pi^*$ satisfies:

$$\pi^* = \operatorname*{argmin}_\pi \sum_{k=1}^K V_k^\pi = \operatorname*{argmin}_\pi \mathring{V}^\pi\,.$$

$\square$

**Lemma 18.** *Choose $c$ to be an arbitrary constant, then we can construct a policy set $\Pi$ for any linear adversarial MDP $\mathcal{M}(\mathcal{S}, \mathcal{A}, H, \{P_h\}_{h=1}^H, \{\ell_k\}_{k=1}^K)$, such that there exists a policy $\pi \in \Pi$, when compared with the global optimal policy, the regret of which is bounded by 1:*

$$\sum_{s=1}^K V_k^\pi - V_k^{\pi^*} \le 1\,.$$

*So that:*

$$\mathrm{Reg}(K) = \sum_{s=1}^K V_k^{\pi_k} - V_k^{\pi^*} = \max_{\pi \in \Pi} \sum_{s=1}^K V_k^\pi - \sum_{k=1}^K V_k^{\pi^*} + \mathrm{Reg}\,(K; \Pi) \le \mathrm{Reg}\,(K; \Pi) + 1\,.$$

*and the size of $\Pi$ is bounded as:*

$$|\Pi| \le \left(1 + 32K^2 H^4 d^{5/2} \log\left(1 + 16HdK\right)\right)^{dH^2}\,,$$

*where $d$ is the dimension of the feature map.*

*Proof.* According to Lemma A.14 in Wagenmaker and Jamieson (2022) that for any linear MDP $\mathring{\mathcal{M}}(\mathcal{S}, \mathcal{A}, H, \{P_h\}_{h=1}^H, \mathring{\ell})$ with fixed reward function $\mathring{\ell}$, we can construct a policy set, that there exists a policy $\pi \in \Pi$, which approximates the best policy of $\mathring{\mathcal{M}}$ with bias $\left|\mathring{V}^\pi - \mathring{V}^*\right| \le \epsilon'$. And the size of the policy set is bounded as:

$$|\Pi| \le \left(1 + \frac{32H^4 d^{5/2} \log\left(1 + 16Hd/\epsilon'\right)}{(\epsilon')^2}\right)^{dH^2}\,. \tag{23}$$

Notice this construction is based entirely on the set of state action features $\phi(s, a)$ and require no information on the loss or reward function. In the adversarial case, we choose $\mathring{\mathcal{M}}$ to be the average MDP denoted in Lemma 17, and we obtain the regret bound of $\pi$ in all the $K$ episodes:

$$\sum_{k=1}^K V_k^{\pi^*} - V_k^\pi = K\left(\mathring{V}^{\pi^*} - \mathring{V}^\pi\right) = K\left(\mathring{V}^* - \mathring{V}^\pi\right) \le K\epsilon'\,. \tag{24}$$

The proof is finished by taking $\epsilon' = 1/K$ in Equation (23).

$\square$

