# OpenReview forum: "Improved Regret Bounds for Linear Adversarial MDPs via Linear Optimization"
_TMLR — Accepted by TMLR_

### Review · Reviewer_G6Ur · 2023-09-16

**Summary Of Contributions:**

This study is centered around the Adversarial Markov Decision Process (MDP) with a linear Q-value function. By leveraging an exploratory assumption, the author introduces innovative algorithms that lead to a substantial enhancement of the previous regret guarantee, reducing it from $O(K^{6/7})$ to $O(K^{4/5})$. Furthermore, if a simulator for the transition is available, the regret guarantee can be pushed even further, achieving an astonishing $O(\sqrt{K})$ improvement.

**Audience:**

Yes

**Broader Impact Concerns:**

I do not find any potential negative societal impact.

**Claims And Evidence:**

Yes

**Requested Changes:**

See weakness.

**Strengths And Weaknesses:**

Strengths:

1. This paper exhibits exceptional clarity in its presentation and is remarkably comprehensible.

2. The proposed novel algorithm represents a significant advancement over the original approach, achieving an impressive $O(K^{4/5})$ regret guarantee without the need for a simulator.

3. The work conducted here performs a direct analysis of linear features, transforming them into a single linear optimization problem. This particular aspect of the research could prove to be of substantial independent interest.

Weaknesses:

1. Regarding the exploratory assumption, it raises questions that the regret does not appear to depend on the minimum eigenvalue $\lambda$. As demonstrated in Table 1, all previous works under similar assumptions demonstrate a clear dependency on $\lambda$. Providing an explanation in the table or theorem would enhance the paper's clarity.

2. The regret guarantee associated with a simulator necessitates an impractical number of simulator calls, namely $O(K^2)$, to achieve an $O(\sqrt{K})$ regret guarantee. This requirement is significantly greater than the number of episodes, K, and appears to be unrealistic.

3. According to Lemma 1, the policy class's size is exponential, rendering Algorithm 1 and Algorithm 2 highly inefficient.

---

> ### Author Response · Authors · 2023-09-19
> **Reply**
>
> We thank the reviewer for providing valuable comments and suggestions. Please find our response below.
>
> -Dependence on $\lambda$:
>
> In Theorem 1, the dependence of $1/\lambda_{\min}$ appears in the term $C_1$, an additive polynomial dependence and thus not reported in Table 1. We will add an explanation after the revision.
>
> -Number of simulator calls
>
> It is worth noting that the number of simulator calls with polynomial dependence on $K$ is standard and acceptable in the literature.
> In Luo et al. (2021), the number of simulator calls is $O(KAH)^{O(H)}$. Compared with this result, our method requires less number of simulators and achieves improved regret bound.
> The methods in Dai et al. (2023) and Sherman et al. (2023) require $\tilde{O}(K^3), \tilde{O}(K^{7/3})$ simulator calls, respectively, which complexities are also larger than ours.
>
> -Computation complexity
>
> We acknowledge that the computational complexity is tied to the policy covering set, which may not be the most efficient approach. However, it is crucial to recognize that accepting a higher computational load can empower the algorithm to handle more intricate situations and yield better theoretical outcomes.
> In particular, when compared to Luo et al. (2021), our method necessitates weaker assumptions, requires fewer simulator queries (if we use one), accommodates a broader range of infinite action spaces, and delivers superior regret bounds.
> Regarding the computation concern, it is important to mention that this kind of dependency is also a standard occurrence in the reduced linear bandit problem, particularly when the cardinality of the decision set is large (Lemma 3.1 in Dani et al., 2007).

---

### Review · Reviewer_mr3v · 2023-10-02

**Summary Of Contributions:**

This paper is about the very challenging problem of regret minimization of (finite-horizon) linear MDPs with adversarial rewards. Minimax optimality is still open in this setting, but this paper provides significant improvements over previous upper bounds.
The proposed algorithm runs an EXP3-like policy with an optimistic value function and a forced-exploration component based on G-optimal design, but the most peculiar part is how the value-function parameters are estimated: using expected (under the state-action visitation probabilities induced by the policy) features, in turn estimated using a known reward-free learning algorithm by Wagenmaker and Jamieson (2022). Depending on whether this "representation learning" phase can be done "for free" (without suffering any regret) or fully online, the overall algorithm suffers $\tilde{O}(\sqrt{K})$ and $\tilde{O}(K^{4/5})$, respectively. The latter result improves over the previously best known upper bound of $\tilde{O}(K^{6/7})$. The $\tilde{O}(\sqrt{K})$ result was achieved before with a generative model, but never with such a weak notion of simulator. Moreover, in both cases, the exploratory assumption is significantly weaker than in previous works and there is no explicit dependence on the minimum eigenvalue of the feature covariance matrix.

**Audience:**

Yes

**Broader Impact Concerns:**

Not necessary.

**Claims And Evidence:**

Yes

**Requested Changes:**

The only critical request is to expand the discussion of the algorithm by Wagenmaker and Jamieson (2022): how it works and how it is employed by your algorithm.

The following are either suggestions to strengthen the work or questions that could be answered as additional discussions or remarks in the paper:

1. I suggest to explain better what kind of simulator you assume to have access to in Section 1. This becomes clear later in the paper, but it is a bit puzzling in Section 1.
2. Calling $\pi*$ "the optimal policy" is slightly misleading in this adversarial setting, where you compare with the best policy in hindsight. Also, how would the result change if we considered an arbitrary comparator policy?
3. You consider the same feature map $\phi$ for all the stages $h\in[H]$. How would the results change if you considered a possibly different feature map $\phi_h$ for each $h$?
4. Computing the exact G-optimal design seems computationally very hard. What happens if you use one of the common approximation methods? How would this approximation error propagate to the regret bounds?
5. If I'm not mistaken, from a purely technical point of view, the "forced exploration" (optimal design) component is only needed to ensure $\eta|\hat{V}|\le 1$ in the exponential-weights proof. I would highlight this in case someone wants to try to remove this component.
6. Is the optimistic bonus only needed to obtain high-probability bounds?
7. In some places, it would be better to remind that we are working under the linear MDP assumption: in the statement of Lemma 1, in the last equation of page 7

Other (minor) remarks:
- In definition 1, $\mu_h$ should be *signed* measures
- The notation $\mathbb{E}_\pi$ should be defined explicitly
- The structure of linear MDPs guarantees that the value functions of *all* policies can be represented by a linear function
- Algorithm 2 would be more clear by separating the two loops over $h$, and by displaying the estimated transition operator in place of its expression in line 6
- In the appendices, Theorem 2 is incorrectly referred to as "Theorem B" in several places.

**Strengths And Weaknesses:**

The paper makes another small step towards minimax optimality for this setting. Given how challenging the problem is, this is understandable, and there are some elements of originality in this work that could help the community to progress even further.

The paper is mostly well written and clear. The theoretical results and their proofs are clear and, as far as I can tell, correct.
The only part that is rather obscure is how the algorithm by Wagenmaker and Jamieson works and is used in the overall algorithm.
The comparison with related works is complete and detailed.

---

### Review · Reviewer_7xbT · 2023-10-27

**Summary Of Contributions:**

This paper considers online learning in adversarial MDP where the loss functions can change arbitrarily over K episodes and the state space can be arbitrarily large and provides new algorithms for which improved guarantees on the regret upper bounds are provided.

**Audience:**

Yes

**Broader Impact Concerns:**

This is a theoretical work, so no concerns.

**Claims And Evidence:**

No

**Requested Changes:**

Could you highlight your technical novelty?

Could you address the questions above?

**Strengths And Weaknesses:**

The authors propose to run the exponential weight algorithm over the cover set of policies, using the loss estimator for each policy combined with an optimistic term. As an initialization step, the algorithm also computes the G-optimal design for each policy. This result can be seen as a non-trivial adaptation of the Exp4.P algorithm to a linear MDP setting. In general, I think the idea of this work allows avoiding the complications appearing in the previous works, as with this approach problematic terms appearing because of using the performance-difference lemma do not appear here. I also think that idea of computing the exploration distribution via G-optimal design fits nicely here. However, I don’t see a potential to make this approach practical, as it has exponential running time at each time step, and also computing the G-optimal design is computationally costly, and the computational difficulty is never discussed in the paper.

Comments:
•	from Definition 1, it is not clear whether the loss is bounded by one or by \sqrt{d}
•	No discussion on how to compute G-optimal design
•	Lemma 1 - never discussed that exponentially large \Pi makes the runtime at each step of the algorithm exponential in parameters
•	 the reference for Lemma 4 doesn’t looks correct, to which result in Freedman (1975) are you referring to? This was presumably taken from [1], also it is unclear why the formulation needs to state this concentration inequality for a fixed V.
•	It is not clear if the inequality in Lemma 5 should hold with some probability.
•	Proof of Lemma 6, last line, bound on loss is missing


[1] High-Probability Regret Bounds for Bandit Online Linear Optimization, Bartlett et. al.

---

### Decision · Action_Editor_9Uqs · 2023-12-05

**Recommendation:** Accept as is

**Comment:**

All reviewers agree that the paper should be accepted. The work is technically solid, and the authors have made some progress on a difficult problem, although optimality remains open. The authors’ algorithmic approach is novel, allowing them to make technical advances over previous approaches. However, there is a concern that this approach may not be along the path towards an efficient algorithm.

**Audience:**

There is a very active community within RL Theory, and this work makes progress for a difficult problem in this area. Experts within this area believe this work will carry some interest.

**Claims And Evidence:**

All reviewers agree that the claims and evidence are there. In particular, the paper is technically sound (it successfully establishes the results it claims).